



# Structure of massively dilatant faults in Iceland: lessons learned from high resolution UAV data

Christopher Weismüller[1], Janos L. Urai[2], Michael Kettermann[2,4], Christoph von Hagke[2,3], Klaus Reicherter[1]

[1]Institute of Neotectonics and Natural Hazards, RWTH Aachen University, Aachen, 52074, Germany, www.nug.rwth-aachen.de
[2]Institute of Structural Geology, Tectonics and Geomechanics, RWTH Aachen University, Aachen, 52074, Germany, www.ged.rwth-aachen.de
[3]Institute of Geology & Paleontology, RWTH Aachen University, Aachen, 52074, Germany, www.emr.rwth-aachen.de
[4]Now at Department of Geodynamics and Sedimentology, University of Vienna, Althanstraße 14, 1090 Vienna, Austria

*Correspondence to*: Christopher Weismüller (c.weismueller@nug.rwth-aachen.de)

## Abstract

Normal faults in basalts develop massive dilatancy up to several tens of meters close to the Earth's surface and show corresponding interactions with groundwater and lava flow. These massively dilatant faults (MDF) are widespread in extensional settings like Iceland or the East African Rift, but their detailed geometry is not well understood, despite their importance for fluid flow in the subsurface, geohazards or geothermal energy. We present a large set of digital elevation models (DEM) of the surface geometries of MDF with 5-15 cm resolution, acquired along the Icelandic Rift zone using unmanned aerial vehicles (UAV). UAV provide a much higher resolution than aerial/satellite imagery and a much better overview than ground-based fieldwork, thus bridging the gap between outcrop scale and regional observations.

Our data present representative outcrops of MDF, formed in basaltic sequences linked to the Mid Ocean Ridge. We acquired photosets of overlapping images along about 20 km of MDF and processed these using photogrammetry to create high resolution DEMs and ortho-rectified images. We use this dataset to map the faults and their damage zones to measure length, opening width and vertical offset of the faults and identify surface tilt in the damage zones. Ground truthing of the data was done by field observations.

Mapped vertical offsets show typical trends of normal fault growth by segment coalescence. However, opening widths in map-view show variations at much higher frequency, caused by segmentation, collapsed relays and tilted blocks. These effects cause a commonly higher than expected ratio of vertical offset and opening width for a steep normal fault at depth.

Based on field observations and the relationships of opening width and vertical offset, we define three endmember morphologies of MDF: (i) dilatant faults with opening width and vertical offset, (ii) tilted blocks (TB), and (iii) opening mode (mode I) fissures. Field observation of normal faults without visible opening invariably shows that these have an opening filled by recent sediment. TB dominated normal faults tend to have a largest opening width with respect to vertical offsets. Fissures



have opening widths up to 15 m with throw below a 2 m threshold. Plotting opening width versus vertical offset of the fractures shows that there is a continuous transition between the endmembers. We conclude that fractures associated with MDF belong to one larger continuum and the three endmembers are thus not necessarily indicative for fracture maturity.

# 1 Introduction

Extensional faults in cohesive rocks can develop massive dilatancy (several tens of meters) at shallow levels in the crust (Abe et al., 2011; Acocella et al., 2003; van Gent et al., 2010; Gudmundsson, 1987a, 1987b; Holland et al., 2006; Kettermann et al., 2015; Opheim and Gudmundsson, 1989; Rowland et al., 2007; Trippanera et al., 2015). These massively dilatant faults (MDF) are common in rift zones such as the Icelandic Rift, the East African Rift or on volcanoes such as Hawaii or the Campi Flegrei, Italy (Acocella et al., 2003; Gudmundsson, 1987b; Martel and Langley, 2006; Rowland et al., 2007; Vitale and Isaia, 2014).

MDF guide the flux of water, magma or hydrocarbons and are therefore of interest for geo-hazard assessment, hydrocarbon exploration, geothermal energy or geodynamics (Crider and Peacock, 2004; Faulkner et al., 2010; Ferrill and Morris, 2003; Grant and Kattenhorn, 2004; Gudmundsson, 1987a; Kettermann et al., 2015, 2016; Rowland et al., 2007). During the past decades, MDF have been studied in the field (Bubeck et al., 2018; Gudmundsson, 1987b, 1987a; Hjartardóttir et al., 2012; Sonnette et al., 2010; Tibaldi et al., 2016; Trippanera et al., 2015), and using analog and numerical models (Abe et al., 2011;

van Gent et al., 2010; Grant and Kattenhorn, 2004; von Hagke et al., 2019; Holland et al., 2006; Kettermann et al., 2015, 2016, 2019; Martel and Langley, 2006; Smart and Ferrill, 2018). In these studies, the surface geometries have been described including tilted blocks (i.e. blocks tilted towards the hanging wall of a fault) and extension fractures. However, many of the observations are based on local measurements considered representative for the regional structures. In this study we investigate the structure and evolution of massively dilatant faults in Iceland (Fig. 1) with the aim to identify and characterize their surface

geometries at the regional scale at centimeter resolution. We achieve this by studying MDF formed in successive lava flows using high resolution data from unmanned aerial vehicle (UAV) based photogrammetry, bridging the gap between outcrop scale and regional observations. This enables us to quantify the geometry of the studied faults at high detail over the entire fault lengths with the ultimate goal to introduce a new classification scheme. To describe different types of discontinuities we use the terminology of Peacock et al. (2016).

**1.1 Geological Background**

  Iceland is a volcanic island in the Atlantic Ocean on the Mid-Atlantic Ridge separating the Eurasian and North American plates. It is linked to a deep mantle plume (Einarsson, 1991; Lawver and Muller, 1994; Vink, 1984; Wolfe et al., 1997), with associated melt production forming the Icelandic shelf (Brandsdóttir et al., 2015) with a local crustal thickness of at least 25 km (Allen et al., 2002). It is located between the Reykjanes Ridge segment in the SW and the Kolbeinsey Ridge in the N (Fig.

2). Maximum horizontal stresses in Iceland are oriented parallel to the rift axes (Ziegler et al., 2016), with a NE-SW trend on the Reykjanes Peninsula Ridge and N-S trend in the North Volcanic Zone. The orientation of the faults, fissures and dikes





follows this trend (Fig. 2; e.g. Grant and Kattenhorn, 2004; Gudmundsson, 1983, 1987a, 1987b; Hjartardóttir et al., 2012; Opheim and Gudmundsson, 1989). The bedrock is mostly of volcanic origin with ages increasing with distance from the rift (Fig. 2). The succession of lava flows with cooling joints, paleo-soils and hyaloclastite result in locally complex mechanical stratigraphy. Faults and fissures crosscutting this mechanically heterogeneous section reactivate the pre-existing cooling joints

close to the surface, leading to complex geometries (Forslund and Gudmundsson, 1991; Gudmundsson, 1987a; Gudmundsson and Bäckström, 1991; Hatton et al., 1994).

## 1.2 Massively Dilatant Faults

Massively Dilatant Faults (MDF) can form close to the surface in cohesive rocks. At depth, dilatant faults (DF) may form in the presence of high fluid pressures. MDF are characterized by distinct geometries such as sub-vertical fault scarps, rotating

blocks and fractures that remain open up to several tens of meters (Acocella et al., 2000; Grant and Kattenhorn, 2004). Existing studies of MDF in outcrops (Holland et al., 2006) and analogue models show that MDF form in layers where the ratio of rock strength to effective stress is sufficiently high (van Gent et al., 2010). With depth, MDF transition to shear faults due to the increase of lithostatic pressure.

Geometries of MDF have been described for several sites in Iceland with respect to fracture length, opening, throw, obliquity

and segment linkage (Acocella et al., 2000; Bonali et al., 2019a; Bubeck et al., 2017; Gudmundsson, 1987b, 1987a; von Hagke et al., 2019; Tentler and Mazzoli, 2005; Trippanera et al., 2015). Relationships of length and opening are complex (Hatton et al., 1994), have largest openings at the fault center (Gudmundsson, 1987a) and smaller opening and throw at shorter lengths (Tentler and Mazzoli, 2005). Correlations of opening and throw are weak, but  show different distributions at fault tips and centers, possibly caused by different stages of fault growth (Gudmundsson, 1987a; Tentler and Mazzoli, 2005). However, a

direct relation of larger vertical offsets and larger dilatancy has been suggested (Acocella et al., 2003; Trippanera et al., 2015). Several models explaining the growth of MDF exist, ranging from (1) upwards propagation of the fault (Grant and Kattenhorn, 2004), which has been linked to dike intrusion (Trippanera et al., 2015) over (2) linkage of shear faults at depth with tensile fractures at the surface (Abe et al., 2011; Hardy, 2013; Vitale and Isaia, 2014) to (3) nucleation at the surface and downward propagation of the fault until tensile failure is no longer possible (Acocella et al., 2003; van Gent et al., 2010). The transition

from tensile to shear fractures can be envisioned as a broad zone reaching down to depths of up to 1500 m, depending on the mechanical stratigraphy of the fractured units, as well as on fault kinematics (von Hagke et al., 2019; Kettermann et al., 2019). In the following we quantify the surface structure of MDF along different faults. We test whether different characteristic surface structures can be distinguished, and whether it is possible to make inferences on their mechanics based on surface analysis only.





### 1.3 Study sites

The sites chosen for this project include the West Volcanic Zone (WVZ) and North Volcanic Zone (NVZ) on Iceland (Fig. 2). The less than 8 Ma old NVZ (Sæmundsson, 1974) is composed of seven volcanic systems, each with a central volcano and associated N-NNE striking faults and fissure swarms (Gudmundsson, 1995; Hjartardóttir et al., 2015; Sæmundsson, 1974).

We selected these sites because they are representative for the variability of faults on Iceland, as they include faults in purely extensional, but also in oblique rift kinematics. They offer the best outcrop conditions with well-defined structures with only minimal vegetation, soil cover or erosion. Associated with the Krafla volcano in the NVZ, the Krafla fissure swarm stretches ca. 40 km south and 50 km north, mostly in postglacial lava flows (Hjartardóttir et al., 2012), with faults and fissures reaching a maximum opening width of 40 m and vertical offsets up to 42 m (Opheim and Gudmundsson, 1989). Two recent rifting

episodes are documented, with eruptions in 1724-1729, (the Mývatn Fires) and 1975 – 84, (Krafla Fires), both accompanied by strong earthquakes and movement along active faults. The horizontal displacement of the rift system during the Krafla Fires was about 8 m, equal to ca. 500 years of plate divergence  (Hollingsworth et al., 2012; Tryggvason, 1994).

The Theistareykir fissure swarm is located in the rift zone within the NVZ and composed of N-S-striking Holocene fissures and normal faults  (Gudmundsson et al., 1993). The westernmost normal fault of the Theistareykir fissure swarm, known as

the Gudfinnugja Fault, connects with the Húsavik-Flatley fault which also offsets the Kolbeinsey Ridge in the North (Fig. 2) (Gudmundsson et al., 1993; Pasquarè Mariotto et al., 2015; Tibaldi et al., 2016). Further interactions of the Theistareykir fissure swarm with the Tjörnes fracture zone have been suggested by (Bonali et al., 2019a) due to variations in strike direction. The Thingvellir fissure swarm is linked to the Pleistocene Hengill volcanic system by the continuity of the faults and the documented ground movement during the last rifting episode in 1789 (Saemundsson and Saemundsson, 1992). The volcanic

system includes ca. 100 associated fissures and faults, of which some reach opening widths > 60 m and vertical offsets of 40 m, representing the largest postglacial structures of Icelandic rift zones  (Gudmundsson, 1987b).  The fissure swarm consists of two Holocene main faults, known as Hrafnagjá and Almannagjá, that envelop lake Thingvallavatn. Almannagjá, which is subject to this study, has been described by Gudmundsson (1987b) as 7.7 km long and locally up to 64 m wide. The southern tip consists of several en échelon extension fractures, which are interpreted to be related to an older weakness, e.g. a Pleistocene

fault underneath (Gudmundsson, 1987b), as further north the fault consists of several parallel fractures and parallel extension fractures at the northern tip (Gudmundsson, 1987b).

The Vogar fissure swarm is located on the NE of the Reykjanes Peninsula in postglacial lava flows. Here, fissures present 75% of the structural discontinuities (Gudmundsson, 1987a). The Vogar fissure swarm has been the scope of several field studies (Clifton, 2003; Grant and Kattenhorn, 2004; Gudmundsson, 1986, 1987a) and a remote sensing study, introducing a post-

coalescence model for fault growth (Villemin and Bergerat, 2013). The geometry of the fractures of the fissure swarm has been characterized as anastomosing or sinuous (Clifton and Kattenhorn, 2006; Clifton and Schlische, 2003; Gudmundsson, 1987a).



## 2 Methods

We combined satellite remote sensing and airborne imagery and used drone imagery to create 3D surface models and digital elevation models (DEMs). These different methods focus on different scales to acquire data, thus making it possible to bridge the gap between cm- and km-scale observations.

### 5    2.1 Satellite-borne data

To identify areas of interest, we used published datasets of satellite and airborne imagery. Google Earth was used in combination with the aerial photographs from Loftmyndir Inc., that are freely accessible via https://www.map.is/base/. As Iceland has been subject to several landscape shaping volcanic eruptions during the time span that is covered by remote sensing data, we included aerial imagery dating back to the 1950s provided by the National Land Survey of Iceland in our preliminary

remote sensing. Some examples of young volcanic eruptions include the Krafla Fires (Hjartardóttir et al., 2012; Tryggvason, 1986) with nine eruptions between 1975 – 1984 (Thordarson and Larsen, 2007), the 2000 eruption of Hekla (Gudmundsson et al., 1992; Rose et al., 2003), or the Holuhraun eruption during 2014 – 2015, the largest Icelandic eruption since more than two centuries (Geiger et al., 2016; Müller et al., 2017; Schmidt et al., 2015). Published mappings of faults and fissures of the NVZ (Hjartardóttir et al., 2012) and the Reykjanes Peninsula (Clifton and Kattenhorn, 2006) were further used to identify possible

sites. TanDEM-X WorldDEM™ tiles with a resolution of 12 m were used at a later stage to complement our own DEMs, which are highly resolved but only able to cover respectively smaller areas. With TanDEM-X WorldDEM™, general surface slopes were identified to quality check our own models in order to avoid typical error sources such as doming (James and Robson, 2014) known to possibly accompany SfM-based DEMs. Furthermore, the elevation models were used to aid and back the interpretations in areas where we do not have coverage with own spatial data.

### 20    2.2 Unmanned aerial vehicle photogrammetry

Unmanned aerial vehicles (UAVs) have been used increasingly in geosciences within the last years, using commercial ready to fly sets or self-made products. They have been applied successfully to e.g. aid the mapping of faults and joints (Bonali et al., 2019b; Vasuki et al., 2014), landslides (Niethammer et al., 2012), or moss beds (Lucieer et al., 2014). The data for this study was acquired using three different UAVs: the DJI Phantom 4 and Mavic pro with 12 MP sensors and the Phantom 4

Advanced, with a 20 MP sensor. We took front- and sideways overlapping photographs of the faults with the cameras facing 90° downwards, covering the area to be mapped, and added oblique photographs to reduce possible 'doming' effects as suggested by (James and Robson, 2014). Due to the large distances we covered, most flights to acquire the photographs were undertaken manually. For later processing in Agisoft Photoscan, we aimed for a frontal overlap of at least 70 % and a sidelap of at least 50 %. We varied the altitude from which the photographs were taken between 30 m and 70 m above ground according

to the estimated accuracy of the resulting DEM and the general dimension of the area to be mapped. We thereby kept the flight





altitudes as low as possible, depending on the dimension of the survey area. The focus of our photographs was set on the fractures and the adjacent areas to capture structures linked to the fracture geometry, such as TB or the damage zone and identify possible surface variations such as topographic slopes. The photographs were sorted according to associated survey areas and reduced to only use sharp photographs with good image quality.

Our method is similar the one used by (Bonali et al., 2019b) who collected data from 50 and 100 m altitude and have shown that with these setting one obtains sufficient resolution for detailed analysis of faults and fissures. We did not place further ground control points (GCPs) and relied on the integrated GPS receiver of the UAV in favor of time efficiency, since absolute elevations are not relevant to our research goals. We used Agisoft Photoscan as processing tool to align the photographs on high settings and create a sparse 3D point cloud. As second step, a dense point cloud was created, that also served as basis for

the DEM, in medium or high resolution, in case the medium setting resulted in DEMs with resolutions > 15 cm/pixel. The resulting DEMs were further used as basis for ortho-mosaics with half the spatial resolution, thus representing the lower threshold of the accuracy of our measurements.

GPS tags by the UAV onboard receiver lead to an absolute horizontal accuracy of ~ 5 m per photograph and a vertical error in the magnitude of several tens of meters. To account for the vertical error of the UAV onboard GPS, we used relative elevations

in our DEMs, assigning 0 m elevation to the respectively lowest point in the model. With the large amounts of photographs taken, the horizontal error is further reduced during the processing, so that we cannot identify a significant error according to the horizontal orientation of our models compared to e.g. TanDEM-X WorldDEM™. To avoid misinterpretations stemming from model tilt, possible slopes and artefacts have been ruled out by comparison with TanDEM-X WorldDEM™ data. To more carefully review the point clouds created during the process and to take first test measurements, we used the Compass

plugin (Thiele et al., 2017) for CloudCompare.

### 2.3 Data extraction

The DEMs and ortho-mosaics were imported in Esri ArcMap to map the fractures manually as polylines on a scale of 1:100. The chosen mapping scale represents the middle ground between a more time consuming, high accuracy or a fast, but low accuracy mapping scale, still enabling us to accurately map details of few tens of centimeters such as the edges of cooling

columns. With a mapping accuracy of a few mm in the DEM and ortho-mosaic at a 1:100 scale, the mapping error is in the same order of magnitude as our spatial resolution. We traced all observable fractures along their surface expressions, including the surface traces of faults, the foot- and hanging wall of DF as well as the adjacent fracture traces of dilatant joints. In a later step, the polylines were merged to polygons, representing the opening of the fracture on the surface. The strike directions of the fractures were measured as represented by a straight line from tip to tip.



### 2.3.1 Opening width and dilatancy

To extract the opening widths, parallel scanlines with 1 m spacing were created orthogonally to the average strike direction. Small deviations in strike were found to have no significant impact on the results, as the relative error is below 1% for deviations < 60° (Supplement S2). The scanlines were subsequently clipped with the polygons of the faults to measure opening

width (e.g. S6). The overall dilatancy of a fault is defined as the summed opening width of subparallel faults and fissures.

### 2.3.2 Vertical offset

Vertical offset was measured on the same scanlines along which the opening width was determined. One measurement point was created along the scanline on the hanging wall and on the footwall (S6). The vertical offset was calculated from the elevation difference and linked to the opening width measurement and the associated scanline. To prevent errors caused by

local variations of surface elevation associated with the damage zone, the elevation data were extracted a few meters away from the fracture, if possible, without crossing another fracture. In this process, we also tried to avoid areas of rough surface with wavelengths of a few meters (large vegetation or lava blocks). To extract geometries of fractures in single models, these methods of opening width and vertical offset extraction are viable. To compare absolute elevations in several DEMs, however, the use of GCP and the correction of the elevations is suggested.

### 2.3.3 Ground truthing and field observations

Ground based observations are used to complement the airborne datasets and interpretations. Vertical fracture walls, which are not as well resolved as horizontal surfaces in the top-down UAV photographs, mostly consist of successive lava flows (Fig. 3, S5). The thickness of the lava flows varies between few cm to meters. Contacts of the lava flows can be smooth transitions

or sharp edges, locally with remains of volcanic glass, mm to a few cm thick. The different morphology of the contacts is interpreted to influence the cohesion between the layers. Patterns of cooling joints may vary strongly between the layers, resulting in columns of few cm to several meters in length and diameter.

The correlation of piercing points of adjacent fracture walls as presented in Bonali et al., (2019b) is more accurate for fissures with small opening width with respect to larger opening widths. More correlation points can be found directly on the fracture

walls. Thus, we experienced the identification of reliable piercing points to be increasingly challenging with increasing opening width and vertical offset. We attribute this to the observation of rubble and disintegrated columns within larger fractures, indicating the advanced erosion of the fracture wall. This is supported by the observations that small fissures may remain unfilled to large depths, while faults are filled with broken columns, typically a few meters under the hanging wall cutoff (Fig. 4b, d). Maximum depths for accessible (larger) fractures and faults observed peak at 30 m up to 50 m in Thingvellir. The true

depth of the opening is not observable, as the voids are filled with rubble, disintegrated columns and sediment. The filling of



the voids can reach up to the fracture edges at the surface, also covering these (Fig. 3a). This leads to the observation of a single scarp with no distinguishable opening from the elevated perspective of an UAV.

The apparent opening width of fissures at the surface can be strongly misestimated when not directly measured at the rock surface: erosion of soil-cover into the fracture leads to funnel shape on the surface, causing overestimation of the opening as

also described in Bonali et al., (2019b). We further observed the reduction of visible opening width by vegetation that i) is large enough to prevent a clear line of sight on the fracture when seen from above and ii) moss and lichen patches that grow over the edges, thus reducing the visible opening (Fig. 5). As the thickness of the moss patches observed can vary between centimeters to decimeters, the relative error becomes larger for overall small opening widths, eventually covering the whole opening on the surface. For opening widths summed over several small fractures, e.g. en échelon arrangements of early stages

of oblique faults, this error will sum up, leading to a large underestimation of the true opening width at the surface.

Ground truthing and quality control of the resulting DEMs was acquired by including reference objects of known dimensions in our models. Our model accuracy lies within the same error range as in Bonali et al., (2019b), who achieved accuracies between 0.04 m – 0.07 m for 50 m - 100 m flight altitude horizontally and up to ca. 20 cm vertically, thus below and within the same order of magnitude as our mapping accuracy.

**3. Results**

**3.1 Digital elevation models**

The combination of our UAV images and the processing software allowed us to create digital elevation models and ortho-rectified mosaics with a resolution varying between 5 and 15 cm per pixel, depending on the i) the altitude the photographs were taken from ii) the count of perspectives achieved per object iii) the quality settings used during the processing. Here, we

focus on several representative DEMs of our dataset that were used to map the faults in detail and to quantify their geometries. A direct comparison between the resolution of our high-resolution DEMs and the ones provided by TanDEM-X WorldDEM™ is provided in Figure 6. To simplify and visualize the fracture geometries, measurements of opening width and vertical offset were sorted according to position along strike (scanline count) and plotted as x-y scatterplot, with the position along strike in meters on the x-axis and the extracted value, (opening width, vertical offset) on the y-axis, scaling with the DEM above (Fig.

25 7-11).

**3.1.1 Asbyrgi**

The field area Asbyrgi is a crack in the graben center (Fig. 7) and is located in the Northern Rift Zone close to the Asbyrgi Canyon tourist spot. The DEM shows a maximum elevation difference of 44 m, that the surface is dipping towards the north at 3°, and a NS-striking fault in the center. A W-E topographic gradient of approximately 5%, with an increased surface dip





towards the E, and locally steeper dips east of the fault can be observed. The size of the DEM covers ca. 1500 m in length and 100 – 150 m width. The fissures in the South (0 – 350 m) are left-stepping en échelon fractures with vertical offsets and opening widths below 2 m. They are underlapping a structure that can be traced from ca. 450 m to 1400 m, consisting of at least 3 segments of similar size, from i) 450 – 700 m, ii) left stepping from 700 – 1100 m and iii) right stepping from 1100 m

– 1400 m. The opening width of the larger structure reaches its maximum values around 10 – 12 m close to its center between 700 and 900 m along strike (al.st.), and decreases towards the tips, with a steeper gradient towards the south than towards the north. The vertical offsets are < 2 m in i) and increase to 3.5 m in ii), where they reach a local minimum in the center, but again increase towards segment iii), in which the vertical offset remains between 2 and 4 m.

### 3.1.2 Krafla North

Figure 8 depicts a N-S-striking fault associated with the Krafla fissure swarm (c.f. Kettermann et al., 2019, their Fig. 2). The DEM covers ca. 800 m x 200 m with an elevation difference of 24 m. The rough surface in the SW is caused by a comparably young lava flow, associated with the Krafla Fires. The center of the DEM shows a ca. 700 m long stretch of a fault which is accompanied by smaller fractures on the hanging wall. Along the main fault, the vertical offset of 15 m remains constant from 150 m towards the end of the DEM in the N, after increasing from 5 m in the S, close to the lava flow. In this section (0 – 150

m al.st.), the vertical offset is underestimated, since the elevations have either to be taken on top of the younger lava flow or on the slope of the TB, which has not been covered by the younger lava. The opening width has a maximum of 17 m and a minimum of 5 m. Sections of large opening width are linked to slopes on the hanging wall, facing away from the footwall, while areas with smaller opening show no significant slope. Starting from 150 m al.st., a breached relay of 100 m length indicates the linkage of two segments, as well as another, smaller lower ramp breach at 580 m.

### 3.1.3 Theistareykir South

The DEM of Theistareykir South (Fig. 9) covers ca. 950 m length and 120 m width with a maximum elevation difference of 39 m. The surface is free of vegetation and has a general slope towards the north. Further trends of dipping surfaces are located on the hanging wall of the fault from 0 – 500 m and 700 m al.st. onwards. The area in between is the lowest area in the DEM. The tips of two overlapping segments of a larger N-S-striking fault structure associated with the Theistareykir fissure swarm

are the essential part in the shown stretch. The fault strike of the single segments slightly varies: N-S from 0 – 450 m al.st., bending ca. 30° towards the E further north. Measurements of the opening width along strike undulate around 12 m in the south (0 – 480 m) and decrease towards the northern tip of the southern segment to ~ 5 m until the northern end of the mapped area. The vertical offset shows less variations with smaller amplitude and larger wavelengths, as compared to the opening width, gently increasing from 4 m in the South towards 6 m at 300 m al.st. Over 50 m distance, the vertical offset then rapidly

increases to its measured maximum of 17 m at 360 m al.st., from where it shows a general decreasing trend towards the north,





interrupted by a local maximum from 640 – 700m al.st. However, in this DEM, a reliable measurements of the vertical offset can only be taken in the central part around 500 m al.st., since a horizontal hanging wall is not covered in the north and the south at the TBs.

### 3.1.4 Thingvellir (Almannagjá)

With almost 7 km length, on average 200 m width and a relative elevation difference up to 53 m, the DEM of the NNE-SSW striking Almannagjá fault in Thingvellir is one of our largest high-resolution data sets (Fig. 10). Bounded by the lake Thingvallavatn in the east, the DEM includes the western main fault of the postglacial graben. It covers the en échelon extension fractures in the south, which connect to larger, segmented fault structures towards the north. The western footwall is characterized by several fault parallel fractures and breached relays, while the hanging wall in the east is accompanied along

strike by an up to 50 m wide, eastwards sloping structure. The measurements of the opening widths are largest at the center of the mapped faults, reaching values up to ~ 64 m, and decline towards the fault tips. Smaller variations in opening width undulate ± 5 m with larger, local maxima in relay zones, e.g. at 1100, 2600, 4000, 5200 m al.st. The vertical offset shows a similar trend: maximum values up to 40 m close to the center of the superordinate fault with decreasing vertical offsets towards the tips in the north and south. Local variations are in the magnitude of few meters, while the general trend is less susceptible

to local undulations. Measurements of vertical offsets in the periphery of 5500 m al.st. are missing, because we were not able to reconstruct a digital elevation model in this area due to an insufficient amount of photographs. Measurements of opening width, however, could be performed based on the ortho-mosaic.

### 3.1.5 Vogar

The last DEM covers the adjacent shoulders of a graben associated with the Vogar fissure swarm (Fig. 11). Our focus while

capturing the drone photographs was on the two NE-SW striking main faults of the graben, however we were able to connect the two photosets by including several traverses orthogonal to the fault strike. Thus, we were also able to cover several smaller fractures in the graben center. The maximum extent of the resulting DEM is more than 2 km in length and ca. 700 m width, neglecting the void areas in the graben center. The relative elevation difference in the DEM is 35 m with the surface sloping towards SW. The northern fracture shows several smaller fractures in the west with opening widths of 1-2 m and vertical

offsets in the range of few decimeters up to two meters (300 – 550 m and 650 – 700 m al.st.). The small, isolated fractures are followed by a larger, connected fault segment from 780 m al.st. on, with opening widths up to 5 m and vertical offsets that increase from < 1 m to several meters towards the east. The following segment from 1200 m al.st. on continues the trend of an increasing vertical offset up to 12 m despite a local minimum at 1400 m al.st. The opening width undulates around 5 m and includes several sections with no measurable opening width, despite a clear vertical offset (e.g. 1400 – 1550 m al.st.). Areas

with a prominent local slope on the hanging wall are located at e.g. 1300, 1650 m, 1700 – 1900 m al.st.

The tip of the southern fracture is not covered in the DEM. Towards the north, the structure continues as several overlapping fractures with relays at 1400, 1600 and 1900 m al.st. The vertical offsets of the southern fracture show less significant changes, with local extrema of 10 m and 3 m, but mainly undulating around 6 m. Opening widths vary strongly, from areas with no measurable openings (e.g. 140 – 210, 310 – 390, 870 – 930 m al.st.) up to locally > 10 m in the first 1000 m al.st. In the
following stretch, the fractures remain open, while still varying strongly between < 1 m and up to 10 m.

**3.2 General observations**

Comparing the four field areas and the observed fracture geometries therein, it is clear, that there are no major differences in structure. Fractures in all areas can be described according to opening width, filled or covered openings, vertical offset and associated structures such as tilted blocks (Fig. 12). In each of the field areas we can define the following endmember structures
from field observations and insights from our DEMs:

1) Fissures

Fissures are opening mode fractures with a prominent surface aperture and no significant vertical offset (Fig. 12b). Their local geometry is governed by pre-existing cooling joints in the basaltic lava flows (Grant and Kattenhorn, 2004; Holland et al.,
2006) and thus develop in dm-scale sawtooth patterns along the boundaries of the basalt columns (Fig. 3a). Fissures can be early stages of MDF or represent the lateral ends of MDF, which show en échelon fracturing when formed at oblique slip (Acocella et al., 2003; Grant and Kattenhorn, 2004; Gudmundsson, 1987a; von Hagke et al., 2019). Viewed in our DEMs (Fig. 13), fissures have a measurable opening width and vertical offset less than 2 m (defined to exceed surface roughness, S1). Earlier studies defined fissures as fractures with vertical offsets < 1 m (Grant and Kattenhorn, 2004; Gudmundsson, 1987b).
Fissures tend to cluster around fault tips, occur in zones parallel to faults, and are rarely longer than 100 m.

2) Dilatant Faults

Dilatant faults, opening mode fractures with vertical walls and measureable opening width and vertical offset (Fig. 12a). The opening can be filled with tilted columns, broken columns, sediment or younger lava flows and vegetation. Basalt columns
can be jammed between the fracture walls or part of the highly porous aggregate between the walls (Fig. 4d). Filling with sediments on top of the rubble can cover the gap completely (Fig. 12c), so that the opening width can no longer be identified in at the surface (Fig. 3a) and DEM (Fig. 14), also described by Trippanera et al. (2015) as their type A. The complete filling of the opening causes the faults to appear as a single scarp on the surface. In basalts, the exposed fracture is sub-vertical and follows the geometry of the basalt columns.
The faults may develop dilatancy of up to 15 m accompanied by vertical offsets in the same order of magnitude. The void between the walls may remain open down to depths of 20 – 30 m, but is usually shallower (Fig. 1). The faces of the fracture walls expose successive layers of lava flows with mostly vertical cooling joints (S5). Field examination of the walls on both





sides usually do not allow a match to be established, indicating that material between the walls is missing due to erosion. In the DEMs (Fig. 15) DF have measurable opening widths, (fracture walls are visible) with clear vertical offsets (Fig. 12a). All faults including the ones that appear non-dilatant can be shown to have an opening which is covered by sediment so that no opening is visible in the DEM (Fig. 14).

3) Tilted Blocks

Tilted blocks (TB), also referred to as monoclines (Grant and Kattenhorn, 2004; Martel and Langley, 2006; Smart and Ferrill, 2018; Sonnette et al., 2010) can develop in the hanging wall of dilatant faults, creating a surface dipping away from the footwall. The length of single tilted blocks ranges from several meters up to several hundred meters; widths range between

several meters to tens of meters, and depending on their subsurface geometry and kinematics three different types of TB can be distinguished (Kettermann et al., 2019).An example of a type III (Kettermann, 2019) TB is provided in Figure 3d. A sketch showing the expected behavior of opening width and vertical offset relationships of a type I TB is provided in Figure 12d. In the DEMs we identify TB quantitatively based on the slope on the hanging wall, dipping away from the footwall (Fig. 16), following Kettermann et al., (2019). Dips of TB are commonly few degrees; exact measurements may be perturbed by

vegetation cover and surface roughness of the lava flows. For a more detailed analysis of TB including kinematic models the reader is referred to Kettermann et al., (2019), here we focus on the presence or absence of TB along the faults.

## 4. Interpretation of the mapped DEM data

We mapped the fractures as previously described and measured opening width and vertical offset along their strike in the

DEMs. Surface structures were identified either in the high-resolution models or, for larger structures, with TanDEM-X WorldDEM™. Based on the definitions above, each measurement in the database was assigned one of the proposed endmember types, i.e. Fissure, Dilatant Fault or Tilted Block.

Asbyrgi (Fig. 17, top):

Due to the small vertical offsets and en échelon arrangement, the fractures in the south are identified as mode I fissures. The larger segmented structure towards the north shows different endmembers: i) is a mode I fracture, followed by two segments of tilted block dominated MDF, as their aspect ratio is biased towards larger opening widths in relation to the vertical offset, further aided by the surface gradient of the hanging wall, which is dipping away from the footwall. As the high-resolution DEM coverage is restricted in these directions, TanDEM-X WorldDEM™ elevations were used to complement the data.





Krafla N (Fig. 17, bottom):

The section of the fault shows and MDF with several TBs. Their direct influence on the measured opening width is apparent in the plot, as the opening width strongly increases along the TB, and decreases again in sections without. The opening width along the breached relay at 150-250 m al.st. has been measured along both overlapping faults and summarized along the

scanline. The cumulative opening width is like the opening width measured in areas without TB, e.g. 500 – 550 m and 650 – 700 m. Thus, the segments are completely merged, because no decreasing opening width is seen, as expected at fault tips. The vertical offset in the south, from 0 m to ca. 190 m along strike is most likely underestimated due to the presence of a TB and the younger lava flow.

Theistareykir S (Fig. 18, top):

The areas along the fault showing the east-dipping surface on the hanging wall (0 – 500 m al.st. and 700 – 950 m al.st.) are interpreted as TBs. The southern TB has a larger opening width when compared with the area between 500 m and 700 m al.st., while the northern T.B. has a larger vertical offset (7 – 10 m) than the southern one, accompanied by opening widths undulating around 6 m. The section between the two TB segments includes a relay zone, with further fractures subparallel to the main

fault. As the cumulative opening widths along the relay zone show no significant variation, the overall extension in this area is interpreted to be the same. Combining an opening width of 5 m with a clear vertical offset and no TB on the hanging wall, the stretch between the TBs in the north and south qualifies as endmember type DF. Furthermore, the decrease in opening width from 450 m al.st. towards the north, coincides with the observed change in strike. With the general orientation of the E-W extension in the northern rift zone and the influence of the Húsavik-Flatley transform fault on the Theistareykir fissure

swarm (Tibaldi et al., 2016), the obliquity of the fault segment is most likely the cause for the decrease in opening width, as proposed by von Hagke et al., (2019).

Vogar, northern fracture (Fig. 18, bottom):

The northern fracture in the DEM of the Vogar fissure swarm represents the northern graben boundary fault and is initiated as

several isolated fractures in the West. Measurements of opening width and vertical offset of these fractures show vertical offsets < 2 m and opening widths up to 6 m. Thus, the fractures including the tip of the larger structure until ca 930 m al.st. are classified as fissures. The following fracture segments towards the east show an overall increasing trend of vertical offset reaching the maximum of 13 m at 1900 m al.st. Reviewing the data from TanDEM-X WorldDEM™, the fracture can be traced further 500 meters from the end of our DEM. Thus, the maximum vertical offset at 930 m al.st. is interpreted as the central

point of the fault ellipsoid. The opening width along strike has a trend similar to the vertical offset, increasing towards the center of the fracture. However, the opening width is less consistent and varies ± 5 m around the general trend over distances of few tens of meters, also resulting in intervals of no measurable opening width, e.g. between 1400 and 1540 m al.st. Several areas with a prominent surface slope, dipping away (SE) from the fracture, can be identified on the hanging wall at 1220-1320,



1700, 1750 – 1950 and 2000 m al.st. and are interpreted as TBs. This structure is a DF with and without tilted block respectively.

Vogar, southern fracture (Fig. 18, bottom):

Partly also presented in von Hagke et al., (2019), the southern fracture consists of several segments that are partially overlapping in their relay zones (e.g. 1000 m, 1400 m, 1900 m al.st.) with vertical offsets consistently fluctuating around 6 m, thus the fractures are classified as DF. Several sections have no measurable opening width (e.g. 140 – 210 m, 310 – 390 m, 430 – 4802 m 870 – 930 m al.st.), because they are filled to the surface, as confirmed by field observation. From 1600 m al.st. towards the NE, the hanging wall surface shows a significant slope towards the graben center. Considering the surface
morphology of the surrounding area, showing a clear rim at the edge of the slope and spatter cones and domes on top of the slope, the slope is most likely caused by a lava flow and not by tectonically induced tilting resulting in TBs. In the context of the local setting, the fault is part of the southern graben boundary fault dipping in the opposite direction as the northern one. Whether this is the actual graben boundary or more an antithetic fault is not clear, as several parallel faults dipping in the same direction can be identified in 400 m and 1000 m SE of the high resolution DEM.

Thingvellir, Almannagjá (Fig. 19):

As described by Gudmundsson (1987b), the Thingvellir fissure swarm consists of two types of fractures: extension fractures with vertical offsets < 0.5 m, in which category most fissures fall, and dilatant normal faults, that often turn into fissures at their tips. Reviewing our DEM, the fault parallel structures along the hanging wall, we interpret the eastwards dipping slope
as several TBs, in accordance with the interpretation of Kettermann et al., (2019). In combination with the large opening width and vertical offset, the proposed endmember type is TB. The en échelon fractures in the southern part of the DEM show vertical offsets < 1 m, when measured on the adjacent sides of single fractures, thus qualify as fissures. However, the eastwards sloping surface is very prominent at this location, resulting in a combined vertical offset of the faults locally exceeding 10 m.

The en échelon fractures have been interpreted as surface expression of an old weakness, possibly a Pleistocene fault that has
been covered by postglacial lava flows (Gudmundsson, 1987b). Thus, the opening width and vertical offsets are interpreted to represent the geometry of a larger, underlying structure. We classify this structure as a very large TB. The overall trends of opening width as well as vertical offset are maximal at the center of the fault and decrease towards the tips. This is interpreted as the typical behavior of an ellipsoidal fault. The dataset of the Almannagjá fault is not only the largest of our datasets with almost 7 km length, but also includes the largest values of opening width, vertical offset and the most prominent TBs. We infer
that these large offsets are possible due to the relatively larger dilation rate of Thingvellir fissure swarm as compared to e.g. the proximate Vogar fissure swarm (Gudmundsson, 1987b).



## 5. Discussion

In this study, we used UAV based photogrammetry to create high resolution DEMs of representative faults and fractures of the Iceland Rift. We show, that these DEMs can be used to map faults and fractures in so far unexcelled detail when compared to past aerial photography or satellite imagery. Furthermore, these DEMs can be used to extract geometries in vast amounts and much faster than taking measurements in field. Resolutions of 5 – 15 cm/pixel are appropriate to map fractures in volcanic settings, corroborating the findings of Bonali et al., (2019b). We derived three endmember types of surface expressions of fractures linked to the massively dilatant faults in Iceland, based on the ratio of opening width and vertical offset, similar to earlier studies on dilatant fractures(Tentler and Mazzoli, 2005; Trippanera et al., 2015, and references therein).

We visualized this ratio for different field areas, color coded for different endmembers (Fig. 20 and Fig. 21). The straight lines plotted in the figure represent the relationship between vertical offset and horizontal opening corresponding to a simple dilatant normal fault at depth with dips ranging between 60°and 70°, being within the dip range as commonly inferred for Iceland (Angelier et al., 1997; Grant and Kattenhorn, 2004; Gudmundsson, 1992; von Hagke et al., 2019; Trippanera et al., 2015). We further define the parameter R as ratio of opening width and vertical offset as Eq.(1):

$$R = \frac{O}{V} \tag{1}$$

where O is the opening width and V the vertical offset measured on one scanline. We calculated R for all endmember types with measured opening width sorted by area (Fig. 22). Expected R values for fault dips between 60° and 70° are within the interval of R(60°) = 0.58 and R(70°) = 0.36.

The fractures in Asbyrgi have been classified as fissures and tilted blocks. The proximity of the fissures to the fault and the prominent surface slope may possibly be interpreted as doming of the surface related to a subsurface dike intrusion, as described for similar structures by Tentler and Temperley (2007). However, in this model, a slope on both sides of the fractures is expected. Since in Asbyrgi the local slope can only be identified at the hanging wall, we interpret the structure as TB. The plot of our data from Asbyrgi does not show a clear separation between the point clouds for fissures and TB (Fig. 20). The difference is a result of our definition for the cutoff of vertical offset (2 m). Thus the data show that TB and fissures are endmembers of a continuum, without a gap or separation in the data. Most measurements from Asbyrgi show much higher R values as would be expected for slip along a basement fault dipping steeper than 60° (Fig. 12 A & D), in agreement with the geometry of TBs (Kettermann et al., 2019) and opening mode fissures.

Perhaps even more interesting are the TB data points with very low opening width in Asbyrgi, because these are not expected for TBs. These measurements are in areas where the visible opening is reduced by vegetation and soil cover. This effect can be seen mostly in relay zones where the opening width is summed over smaller fractures or at the fault tip in the north, thus





indicating an underestimation of opening width by mapping errors. A general trend of the surface dipping at approximately 5% from W to E can be identified in our high resolution and the TanDEM-X WorldDEM™ data. Since the vertical offsets have been measured at a distance to the fracture, an additional error is made, leading to the overestimation of vertical offsets and thus underestimation of R.

Krafla N includes the two endmember types of dilatant faults (DF), with and without TB, which form two clusters (Fig. 20). DF with the lowest R ≈ 0.49 are interpreted to be the closest to "classic DF" (Fig. 12A), but most of the data show much larger R with TB often having R > 1 (Fig. 20). However, the two clusters of R in Krafla overlap. R in data of Theistareykir S show a similar trend, where DF are within the commonly assumed interval of R with R ≈ 0.47, while TB tend towards larger R with
R ≈ 1.07. However, in Theistareykir S, the vertical offsets are underestimated at the TBs due to the missing flat lying part of the hanging wall in the DEM, leading to a too high R value. Overlaps of DF and TB can be explained by the resolution of surface tilt; TB with surface dips << 5° may be unrecognized and misinterpreted as DF. Particularly the transition from TB to DF is prone to interpretation errors when no clear boundary is visible.

The tendency of TB towards large opening width with respectively smaller vertical offset becomes even more apparent in Thingvellir, where TB are well developed and the majority of the measurements have R > 1 (median R ≈ 1.59). The smaller, detached cluster between 0 - 5 m vertical offset and 5 – 20 m opening width (Fig. 20) results from the measurements taken on the en échelon fractures in the southern part of the main fault. These classify as fissures when viewed as single fractures (S 3), but when counted cumulatively, reflect the underlying fault structure as a TB. This strain partitioning has been described as
separate fault structure by Trippanera et al. (2015).
The fractures mapped in Vogar are particularly interesting, as all proposed endmember types are present: fissures form a continuum with DF in the north (Fig. 20), as in Asbyrgi. Faults with no distinguishable opening width were all confirmed in the field to be filled with sediment at the northern and southern fracture. R values of DF in the north (R ≈ 0.65) and south (R ≈ 0.62) are similar to R values of TB (R ≈ 0.67), while TB trend towards larger vertical offsets (Fig. 20). The high value of R
for DF is most likely caused by strong erosion of the fracture walls, leading to overestimation of the opening width, and accumulation of material in the opening, up to completely covering it. With our selected areas all types of TB as defined in Kettermann et al. (2019) are covered: Type I in Vogar and Krafla, Type II in Theistareykir and Type III in Thingvellir, all resulting in R values larger than expected for the "classic" DF. Solely relying on measurements of opening width and vertical offset, thus R, different types of TBs cannot be distinguished. This is because vertical offset is less influenced by surface
structures as the opening width and measured outside the influence area of TB. Additionally, opening width can vary strongly over short distances. Generally, in all areas the data of the endmember classifications form one continuous cloud.

We added further measurements from Asbyrgi and Krafla (S4) in combined plots (Fig. 21) including data from Iceland and Ethiopia (Tentler and Mazzoli, 2005; Trippanera et al., 2015, and references therein). The distribution of the endmember types





remains consistent. However the data of Tentler and Mazzoli, (2005) add a number of measurements with V= 10 – 15 m and O < 5 m, suggesting basement fault dips steeper than 70°. From Figure 21 we infer the following:

i)   Fissures have vertical offsets < 2 m per definition and can accumulate up to 15 m of opening width. Fractures with larger opening widths will also develop larger vertical offsets and thus no longer qualify as fissures. However, there

is no clear correlation between V and O in any of the data sets. Some weak correlation may possibly be inferred for Vogar or Krafla (Fig. 20).

ii)   DF clusters tend to have larger vertical offset than opening width. The net amount of vertical offset and opening width are directly linked to the dip of the basement fault. A basement fault dipping > 45° produces a vertical offset at the surface that is larger than the opening width. DF with V > 10 m concentrate (with some outliers) between R associated

with 60° - 70° basement fault dip. This is in line with the results from analog models by von Hagke et al., (2019), who use a prescribed basement fault dip of 60°. Similarly this fits with the transition from pure extension to a steep normal fault as proposed by Acocella et al. (2003) and Gudmundsson (1992). Measurements classified as DF that show higher values of R (associated with shallower basement fault dips) are the result of erosion and the disintegration of the fracture walls, leading to an overestimation of the opening width.

iii)   TB plot in clusters that also overlap with DF, but mainly have R > 0.5, depending on fault geometry. Faults with large vertical offsets in relation to the opening width can produce TB clusters similar to those of DF, as observed in Vogar. However, when compared to their non TB counterparts, the TB clusters trend towards larger opening widths (S4.5). This trend can be explained by the rotation of the hanging wall away from the footwall and a resulting increased aperture (Kettermann et al., 2019). Therefore, measurements of opening width on TBs lead to an overestimation of

the overall dilatancy of the fractures.

Vertically elongated clusters in the plots (Fig. 21) are the result of relatively stable vertical offsets with smooth gradients over long distances al.st. at most faults, while the opening width shows much more local variations. Piercing point correlations of adjacent fracture walls are more reliable for small openings and opening width measurements can vary strongly over the fault length. Consequently, errors increase towards the fault center where displacement is largest, as shown for Asbyrgi, Krafla and

Theistareykir. Maximum opening widths are larger than 60 m in the Thingvellir dataset with a maximum vertical offset of 40 m. Our data is consistent with the measurements of Tentler and Mazzoli (2005) and Trippanera et al. (2015), who have however not studied faults with large values as present in Thingvellir. Figure 22 shows the distributions of R for the different endmembers, but the overlapping distributions indicate that all mapped structures belong to a larger continuum with smooth transitions between endmembers. Furthermore, vertical offsets with small opening widths are rare in our and the data of Tentler

and Mazzoli (2005) and Trippanera et al. (2015), resulting in a gap between DF with and without TB, and DF with no opening (Fig. 21). The reasons for this gap in our data is i) the mapping procedure of the fracture traces (S6) at the transition between DF and no opening, and ii) the interval length between scanlines (S6). The transition between opening and non-opening occurs over shorter lengths, therefore the transition between opening and the fully filled state with decreasing opening widths is not fully covered by the scanlines.



## 6. Conclusions

From the measurements and interpretations, we derive that:

- Measurements of vertical offset follow the trend of an elliptical fault without much local variation, whereas opening width is more prone to local variations when measured along strike.

- The local variations in opening width can be caused by formation of tilted blocks or by erosion, corroborating earlier studies. Erosional processes such as collapse of the fracture walls or disintegrated relays of the fracture walls may lead to overestimation of opening width; when fractures are filled and/or covered by sediment or vegetation, opening width may be underestimated

- Structures that appear as non-dilatant normal faults on the surface can consistently be shown to have a blind opening
10 hidden by vegetation and sedimentation.

- Tilted blocks are common features observed along all faults. They may be present along the entire fault (Thingvellir), or absent over several kilometers (Vogar S).

- Underestimation of the vertical offset of the master fault at depth occurs when measurements are taken on the slope of a tilted block.

- Structural endmembers (Fissures, Dilatant Faults, Tilted Blocks) are part of a continuum with smooth transitions. Based on measurements of horizontal and vertical offset only, it is not possible to infer fracture mechanics.

- Vertical offset is controlled by deep processes, while opening widths are influenced by surface processes

Results of this study can be used in the future to validate scaled analogue or numerical models in order to better predict MDF
20 structures at depth. This may help guiding geothermal or hydrocarbon exploration.


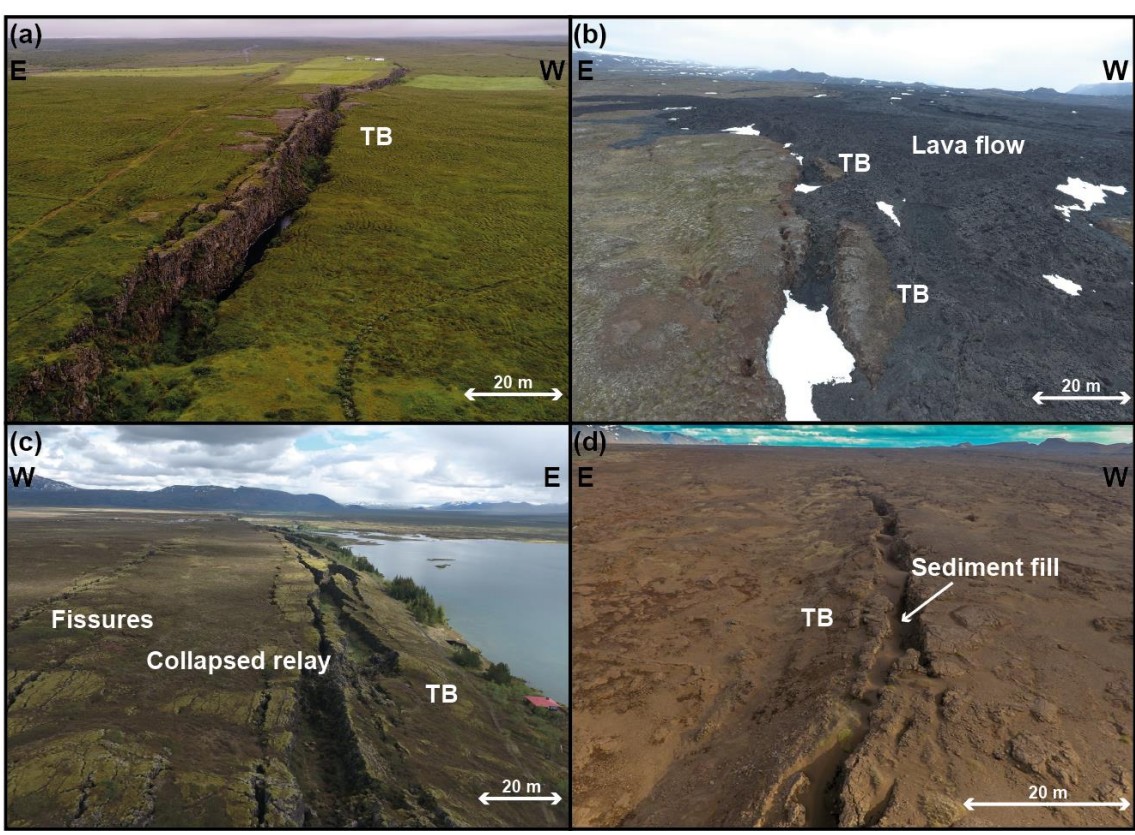

**Figure 1:** (a) MDF close to the Asbyrgi canyon. Along this fault segment, opening widths reach up to 20 m and vertical offsets up to ca. 15 m of displacement. The void between footwall (E) and hanging wall block (W) is partially filled with rubble or water. Soil with earth hummocks or thúfa, which are characteristic for periglacial environments, covers the surface and possible smaller fractures associated with the main fault, TB = tilted block. 16°34'15"W 66°2'14"N (b) MDF of the Krafla fissure swarm. The hanging wall has been covered by a lava flow from the Krafla Fires, also surrounding the tilted blocks. 16°44'32"W 65°48'44"N (c) Southern part of the Almannagjá fault in Thingvellir with lake Thingvallavatn on the right. A prominent tilted block is dipping eastwards and fault segments are linked by a collapsed relay. Fissures in the W. 21°8'30"W 64°14'42"N (d) Fault segment of the Theistareykir fissure swarm. The opening of the dilatant fault (footwall in the W) with an eastwards dipping tilted block is being filled by sands and aeolian sediments. 17°0'20"W 65°50'50.38"N.





**Figure 2:** (a) Geological overview of Iceland, simplified from (1989) . Jarðfræðikort af Íslandi - Berggrunnur - 1:600.000 - NI_J600v_berg_2.utg. https://gatt.lmi.is:geonetwork/srv/api/records/{005FFDAD-69A1-4385-B16F-FD31B960FE33}. Rift zones as introduced in (Thordarson and Larsen, 2007) and the study areas with the surveyed faults are indicated. EVZ: East Volcanic Zone; KR: Kolbeinsey Ridge; NVZ: North Volcanic Zone; RR: Reykjanes Ridge; TFZ: Tjörnes Fracture Zone WVZ: West Volcanic Zone. (b) Detailed view of the Reykjanes Peninsula and West Volcanic Zone, the presented faults are taken from (Clifton and Kattenhorn, 2006). (c) Detailed view of the geology and study areas in the North Volcanic Zone. The mapped faults are taken from (Hjartardóttir et al., 2012). Projection: WGS1984 UTM 27N and 28N.





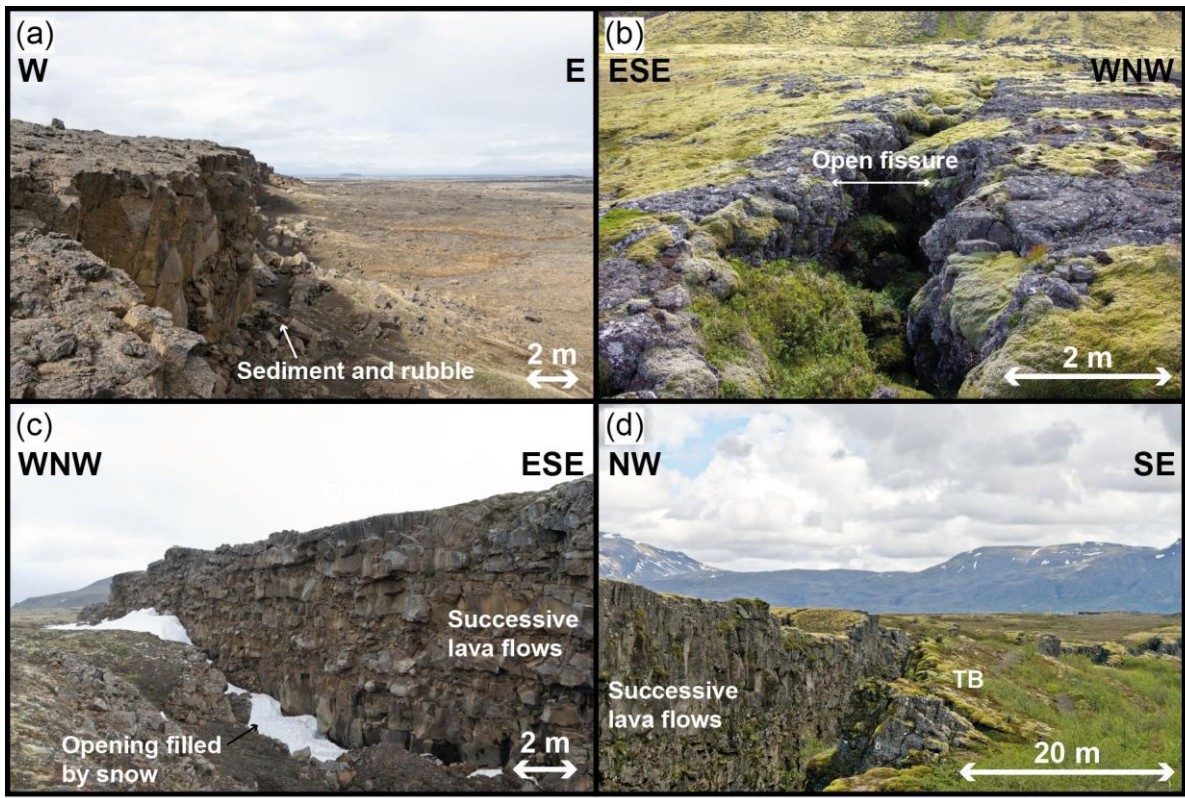

**Figure 3:** (a) Normal fault with no visible opening width but a clear scarp with vertical offset. Aeolian sediments and rubble form at the bottom of the scarp, forming a colluvial wedge. 16°17'27"W 65°32'12"N (b) Fissure with no vertical offset but a prominent opening width. 21°43'20"W 64°00'11"N (c) Dilatant fault, combining opening width with vertical offset. The free face of the fracture shows a succession of lava flows. 16°44'3"W 65°48'43"N (d) Tilted block on a dilatant fault, note the slope on the right side of the image dipping away from the main fault. 21°8'37"W 64°14'33"N.





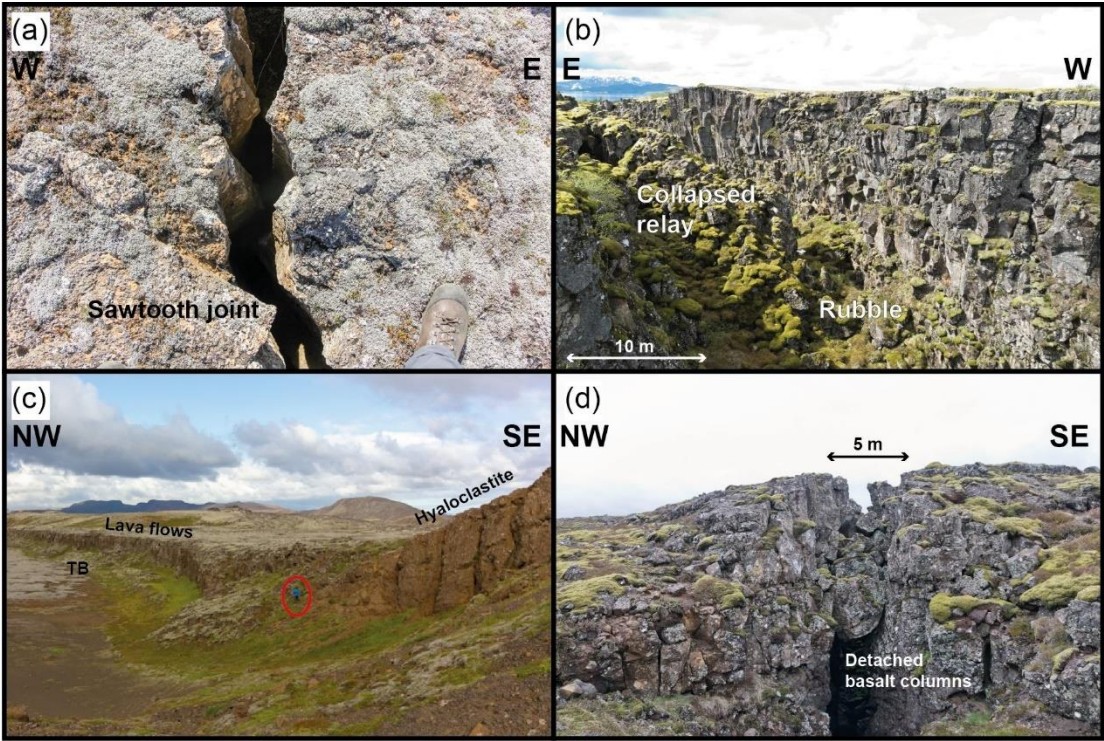

**Figure 4:** (a) Cooling joint with the typical sawtooth pattern. 16°37'48"W 66°2"55"N (b) MDF in Thingvellir. The relay has collapsed, filling the opening of the fault with rubble 21°8'38"W 64°14'34"N (c) A fault that dissects different material developed tilted blocks: Lava flows on the left and softer hyaloclastite on the right. Person for scale. 22°11'41"W 63°51'48"N  (d) Side view inside a dilatant fault. Note the detached basalt columns that are stuck between the fracture walls, preventing the cavity beneath from being filled by further rubble from above. 22°20'13"W 63°57'58'N.




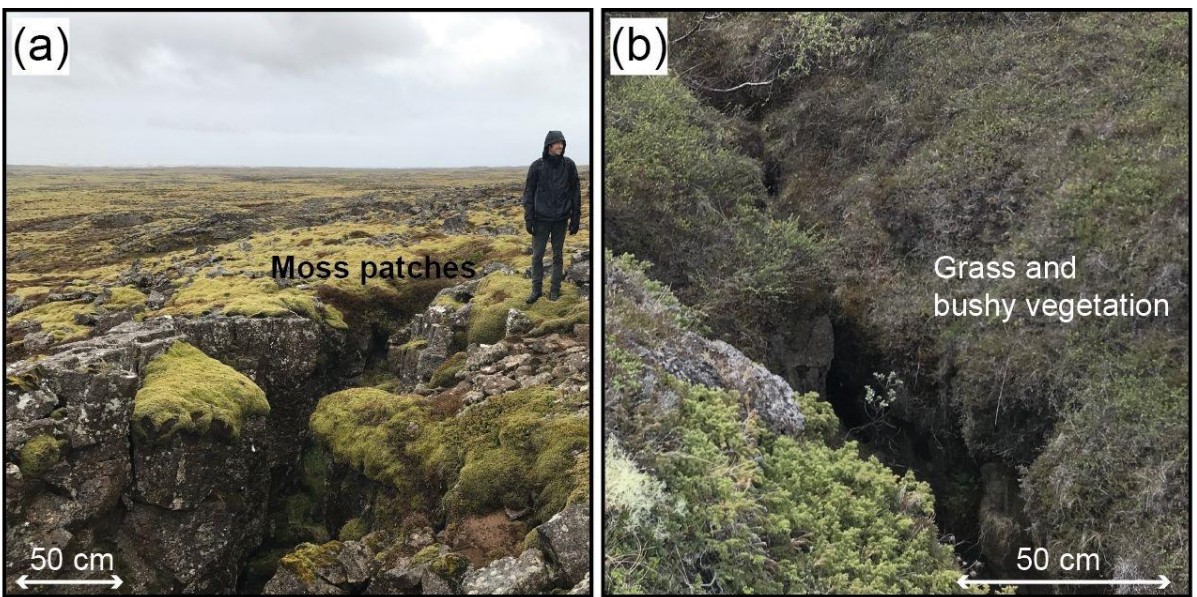

**Figure 5:** (a) Moss patches have grown over the edges of the fractures, reducing the visible opening locally by the thickness of the moss patch (front) and also completely as visible left of the person. 22°33'41"W 63°54'27"N (b) Grass and bushy vegetation can also cover the edges of fractures in areas with soil on top of the bedrock, reducing the visible opening width when viewed from top and impair a clear line of sight from an UAV. 16°35'13"W 66°1'24"N.

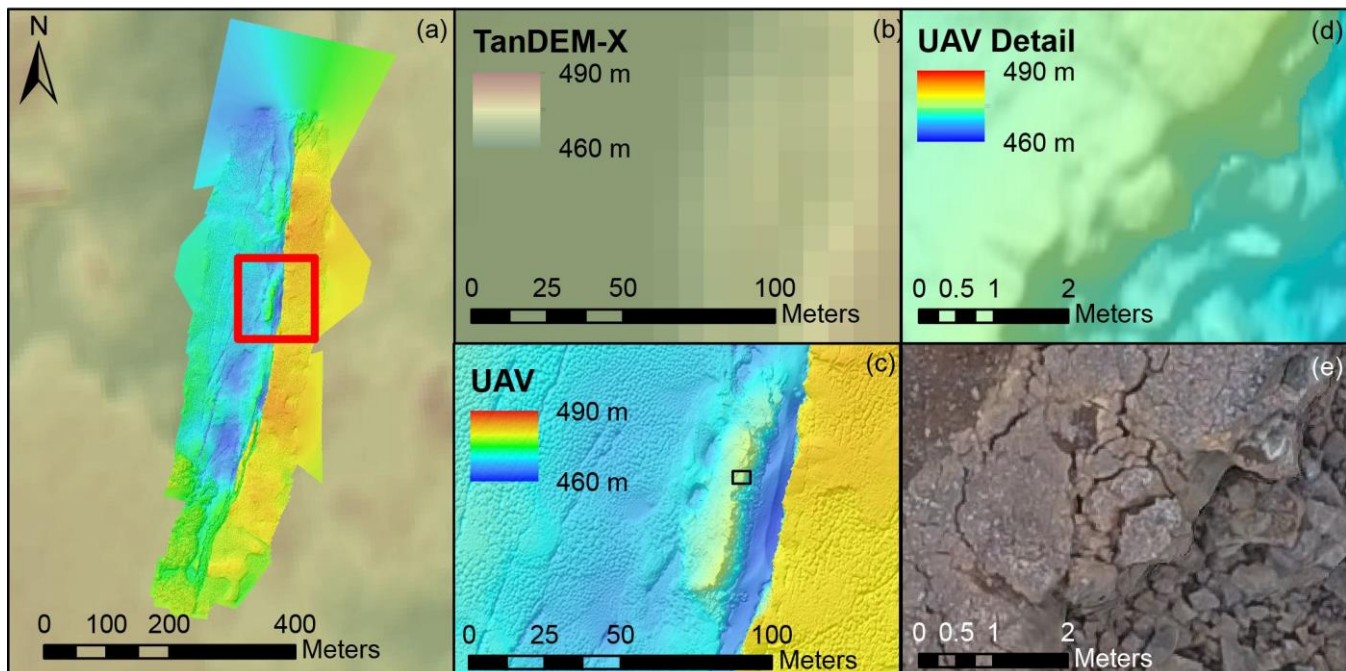

**Figure 6:** (a) Drone based high-resolution DEM of fault belonging to the Krafla fissure swarm, projected on TanDEM-X WorldDEM™. (b) Detailed view on the TanDEM-X WorldDEM™ tile of the area marked in a. (c) High-resolution DEM analogue to b. (d) Detailed view of



area marked in c of the high resolution DEM. (e) Ortho-rectified UAV-photography analogue to d. Location of the DEM: 16°43'12"W 65°51'19"N. Projection of the DEM and ortho-mosaics: WGS1984 UTM 28N.

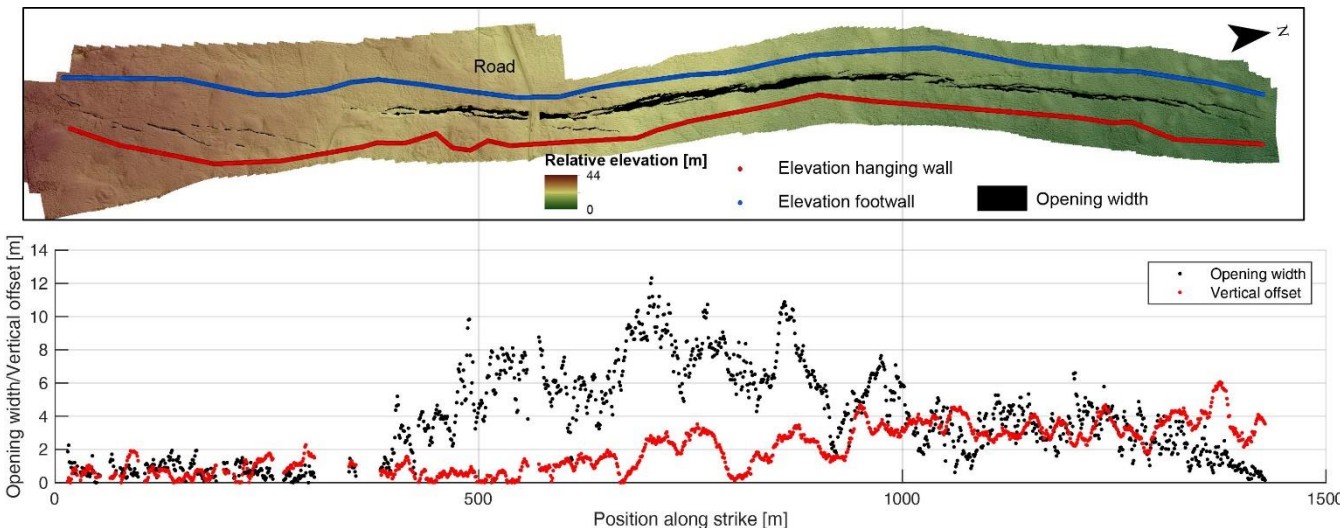

5 **Figure 7:** Fracture close to the Asbyrgi canyon. Top: DEM with the mapped opening width of the fractures and elevation extraction points (blue points on hanging wall, red points on footwall; points were selected on a line placed such that local topographic variations are avoided or at a minimum). Bottom: Opening width (black) and vertical offset (red) plotted along strike. The x-axis scale is analogue to the scale in the DEM on top. 16°35'14"W 66°01'21"N. Projection of the DEM: WGS1984 UTM 28N.




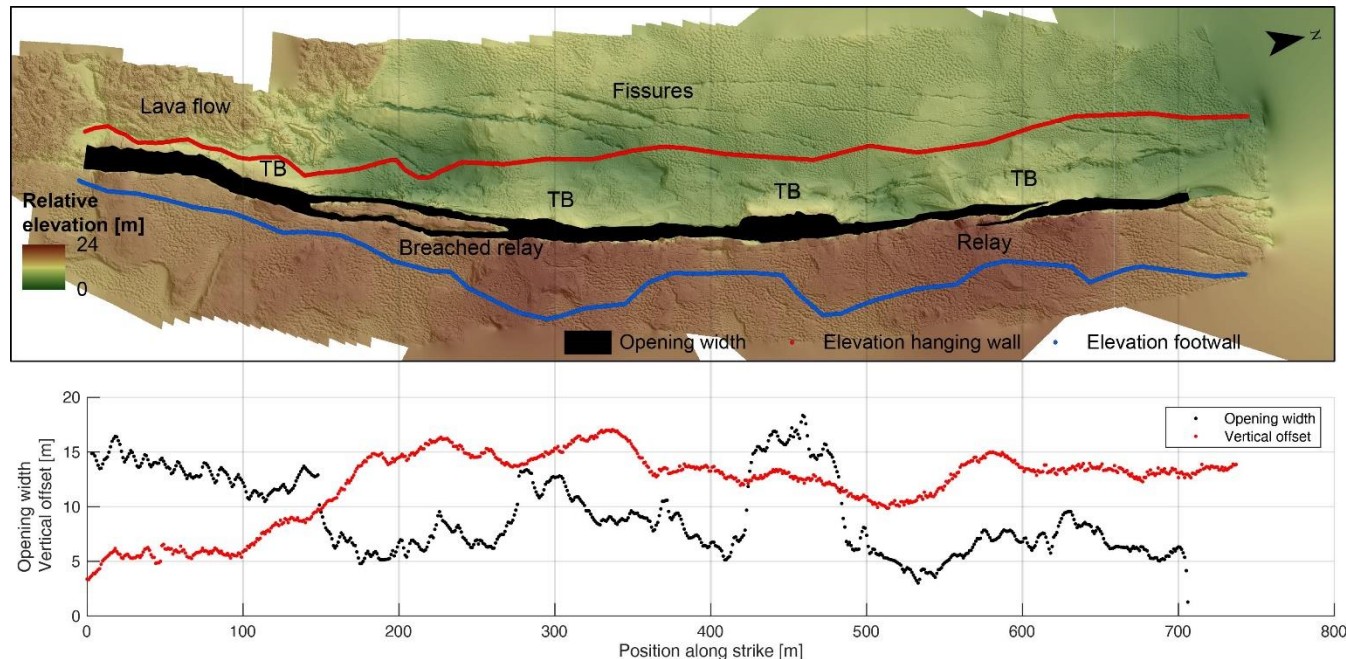

**Figure 8:** Fracture associated with the Krafla fissure swarm. Red dots: elevation extraction points on the hanging wall, blue dots: elevation extraction points on the foot wall. Top: DEM with the mapped opening width of the fractures and elevation extraction points. Bottom: Opening width (black) and vertical offset (red) plotted along strike. The x-axis scale is analogue to the scale in the DEM on top. 16°43'21"W 65°51'19"N. Projection of the DEM: WGS1984 UTM 28N.

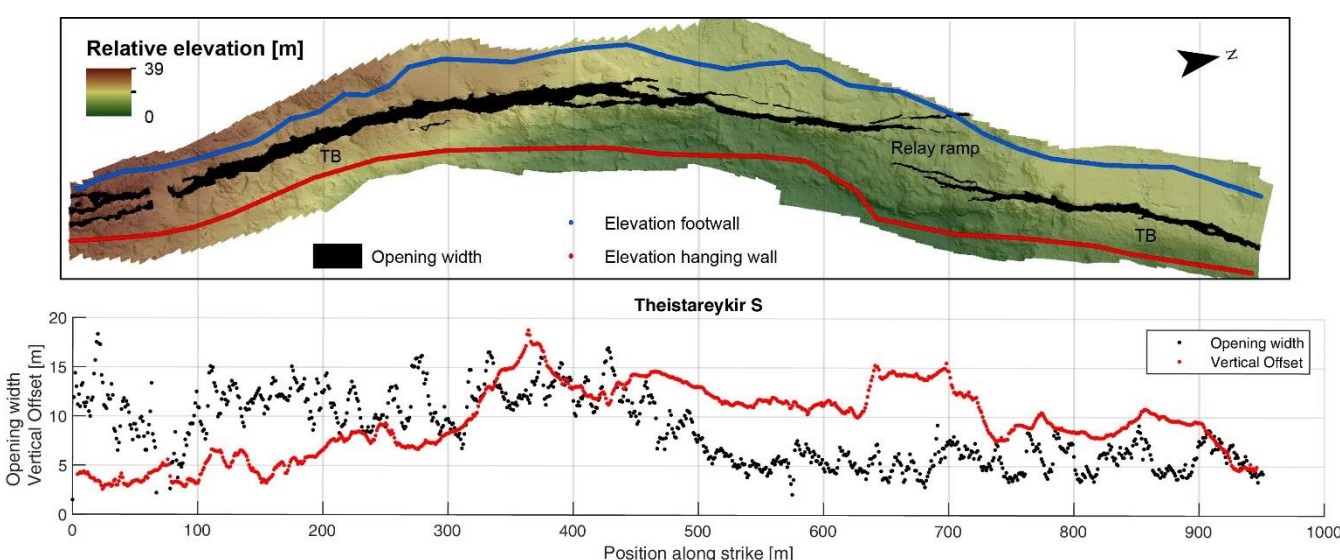

**Figure 9:** Fracture associated with the Theistareykir fissure swarm. Top: DEM with the mapped opening width of the fractures and elevation extraction points. Bottom: Opening width (black) and vertical offset (red) plotted along strike. The x-axis scale is analogue to the scale in the DEM on top. 17°00'23"W 65°50'45"N. Projection of the DEM: WGS1984 UTM 28N.





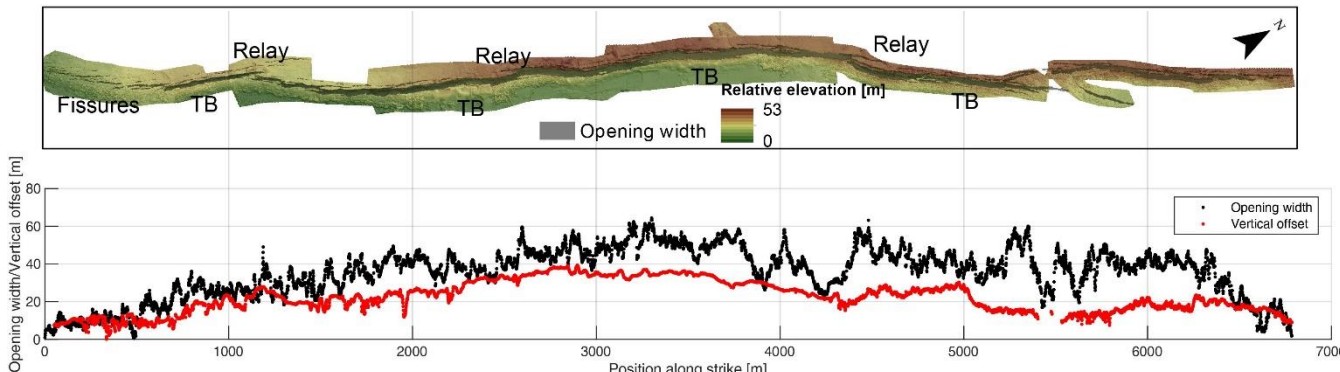

**Figure 10:** Fracture associated with the Thingvellir (Hengill) fissure swarm, also known as the Almannagjá fault. Top: DEM with the mapped opening width of the fractures. Due to the large scale of the model, elevation extraction points have not been plotted. Bottom: Opening width (black) and vertical offset (red) plotted along strike. The x-axis scale is analogue to the scale in the DEM on top. 21°8'37"W 64°14'33"N. Projection of the DEM: WGS1984 UTM 27N.





**Figure 11:** Fractures of the Vogar fissure swarm. Top: DEM with the mapped opening width of the fractures and elevation extraction points. The elevation extraction points of the fissures in the NW are not shown due to the scale. Middle: Opening width (black) and vertical offset (red) plotted along strike of the northern fracture. Bottom: Opening width (black) and vertical offset (red) plotted along strike of the southern fracture. The x-axis scale of both plots is analogue to the scale in the DEM on top. 22°20'40"W 63°58'0"N. Projection of the DEM: WGS1984 UTM 27N.





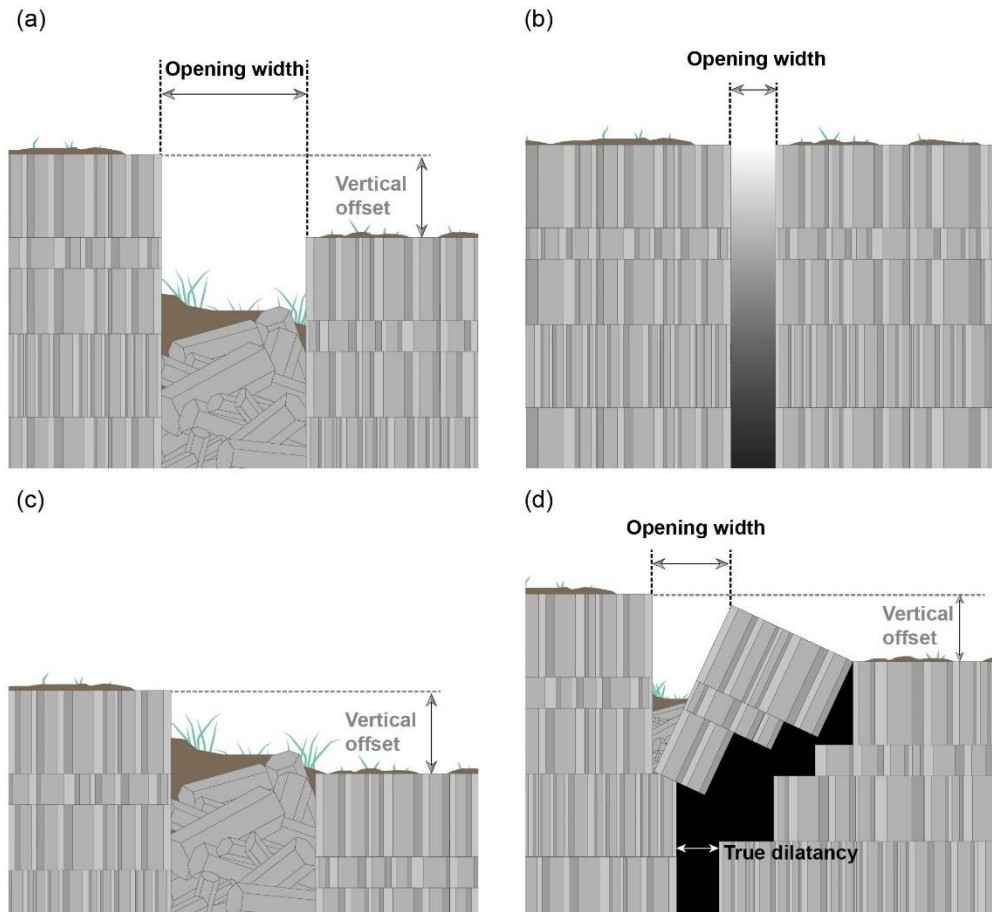

**Figure 12:** (a) Dilatant fault. The opening is filled by detached columns, rubble and sediment. The opening width is measured from edge to edge, vertical offset from the top of the footwall to the top of the hanging wall with a distance to the opening. (b) Fissure, the opening is measured from edge to edge and can remain open (unfilled) to uncertain depth. (c) Dilatant fault with the opening filled to the surface, thus preventing accurate measurements. Vertical offset is measured from the top of the footwall to the top of the hanging wall. (d) Tilted block (Type I of Kettermann et al., 2019). The opening width is measured from edge to edge and can be filled with detached columns, rubble or sediment. The opening on the surface is larger than the true dilatancy at depth. Vertical offset is measured from the top of the footwall to the top of the hanging wall with a distance to the TB. Not to scale.




**Figure 13:** Samples showing the resolution of our DEMs and ortho-mosaics on fissures, left: ortho-rectified mosaics, right: DEM. (a) - (d) Asbyrgi, strong vegetation and soil cover the clear edges of the fissures. Erosion of soil into the fissure leads to a funnel shaped opening at the surface. 16°35'20"W 66°1'8"N and 16°35'19"W 66°1'10"N (e),(f) Vogar, where less soil and vegetation allow a more accurate mapping of the opening. 22°21'59"W 63°57'55"N. Projection: WGS1984 UTM 27N and 28N.



**Figure 14:** Samples showing the resolution of our DEMs and ortho-mosaics on dilatant faults with filled openings in Vogar, left: ortho-rectified mosaics, right: DEM. (a), (b) Vogar, the opening is still visible in the South, but  complete filled by basalt rubble in the north of the image. 22°21'47"W 63°57'30"N (c) – (f): Rubble and vegetation covering the openings. 22°20'59"W 63°58'11"N and 22°20'52"W 63°58'12"N. Projection: WGS1984 UTM 27N and 28N.



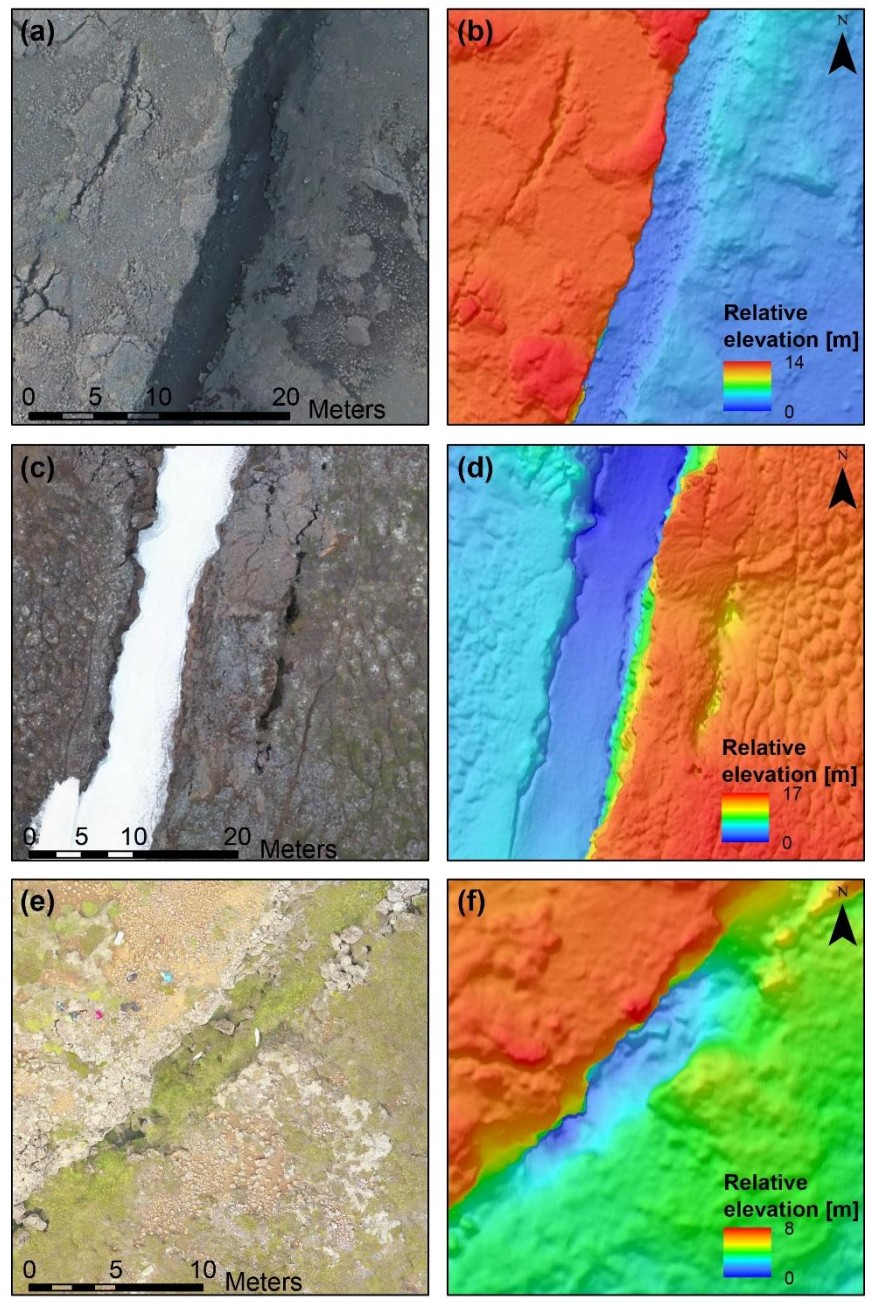

**Figure 15:** Samples showing the resolution of our DEMs and ortho-mosaics on dilatant faults, left: ortho-rectified mosaics, right: DEM. (a), (b) Theistareykir, the opening is not yet completely filled by sediment. 17°0'19"W 65°50'50"N (c), (d) Krafla, the opening is partly and temporarily filled by ice. 16°43'23"W 65°51'17"N (e), (f) Vogar, the opening is partly filed by rubble and vegetation, strong erosion is apparent. 22°21'16"W 63°58'5"N. Projection: WGS1984 UTM 27N and 28N.





**Figure 16:** Samples showing the resolution of our DEMs and ortho-mosaics on tilted blocks, left: ortho-rectified mosaics, right: DEM. (a), (b) Krafla, the opening is partly filled by snow. The surface dip of the TB is visible in the DEM. 16°43'23W 65°51'19"N (c), (d) Partly segmented TB in Thingvellir, including the Öxarárfoss waterfall. 21°7'4"W 64°15'56"N (e), (f) TB dipping towards the South in Vogar. 22°20'29"W 63°58'19"N. Projection: WGS1984 UTM 27N and 28N.





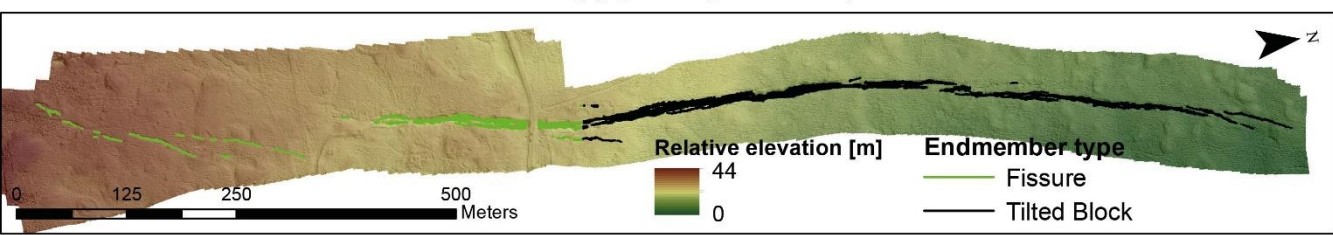

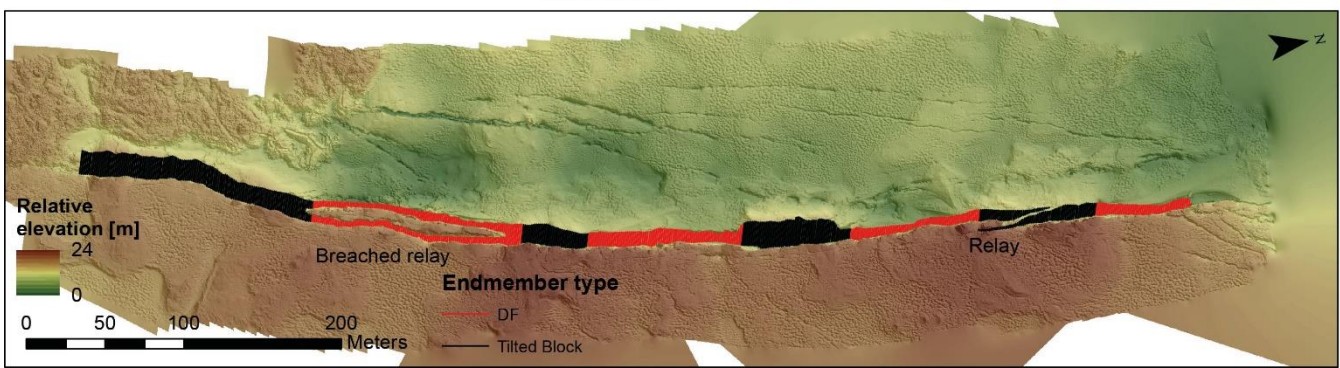

**Figure 17:** Top: DEM and proposed endmember types of the fractures close to Asbyrgi canyon. Bottom: DEM and proposed endmember types of the fractures of the Krafla fissure swarm. Projection of the DEM: WGS1984 UTM 28N.





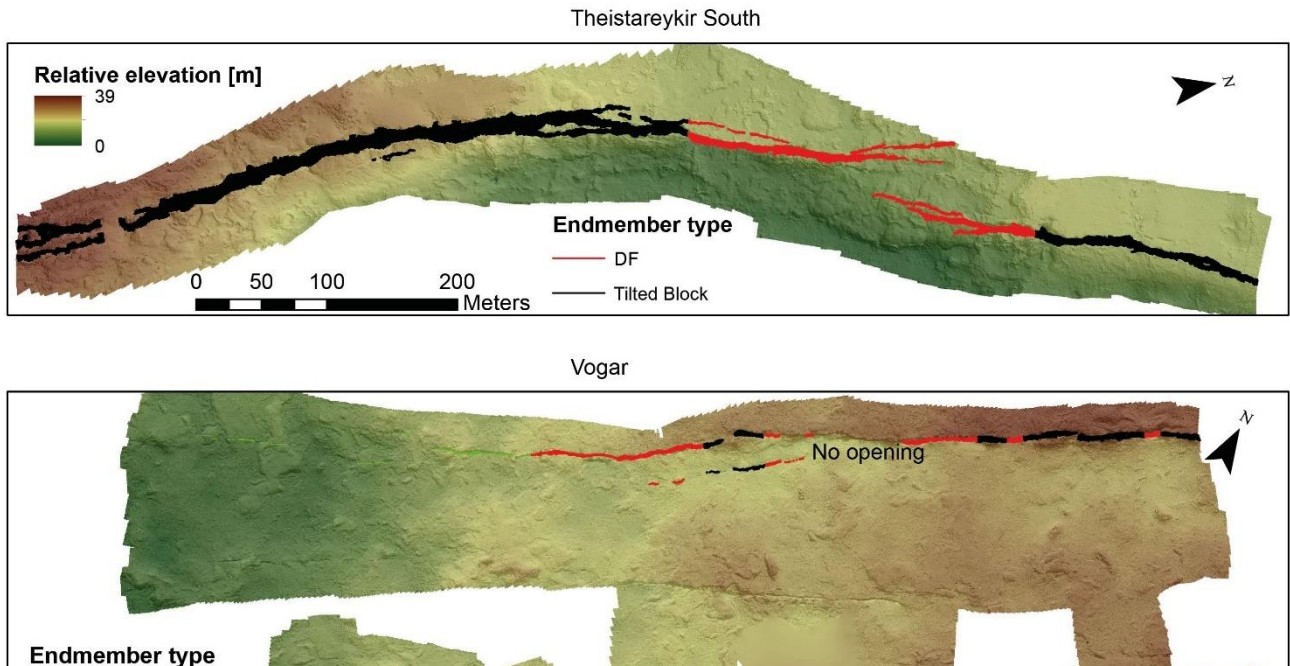

**Figure 18:** Top: DEM and proposed endmember types of the fractures of the Theistareykir fissure swarm. Bottom: DEM and proposed endmember types of the fractures of the Vogar fissure swarm. Projection of the DEM: WGS1984 UTM 28N (top) and UTM 27N (bottom).

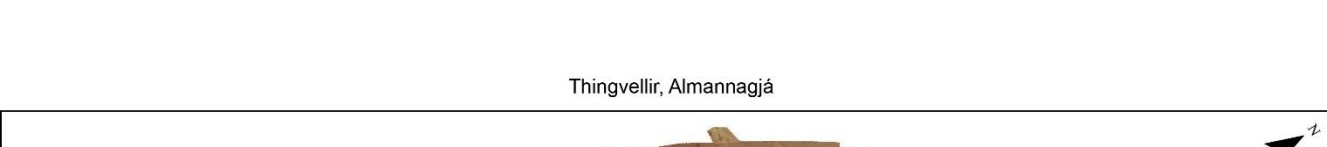

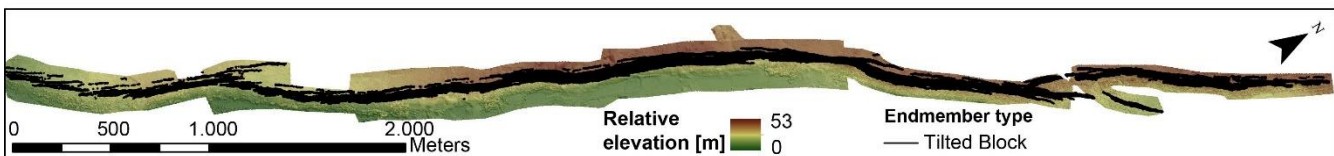

**Figure 19:** DEM and proposed endmember types of the fractures of the Thingvellir fissure swarm. Projection of the DEM: WGS1984 UTM 27N.

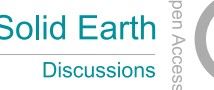

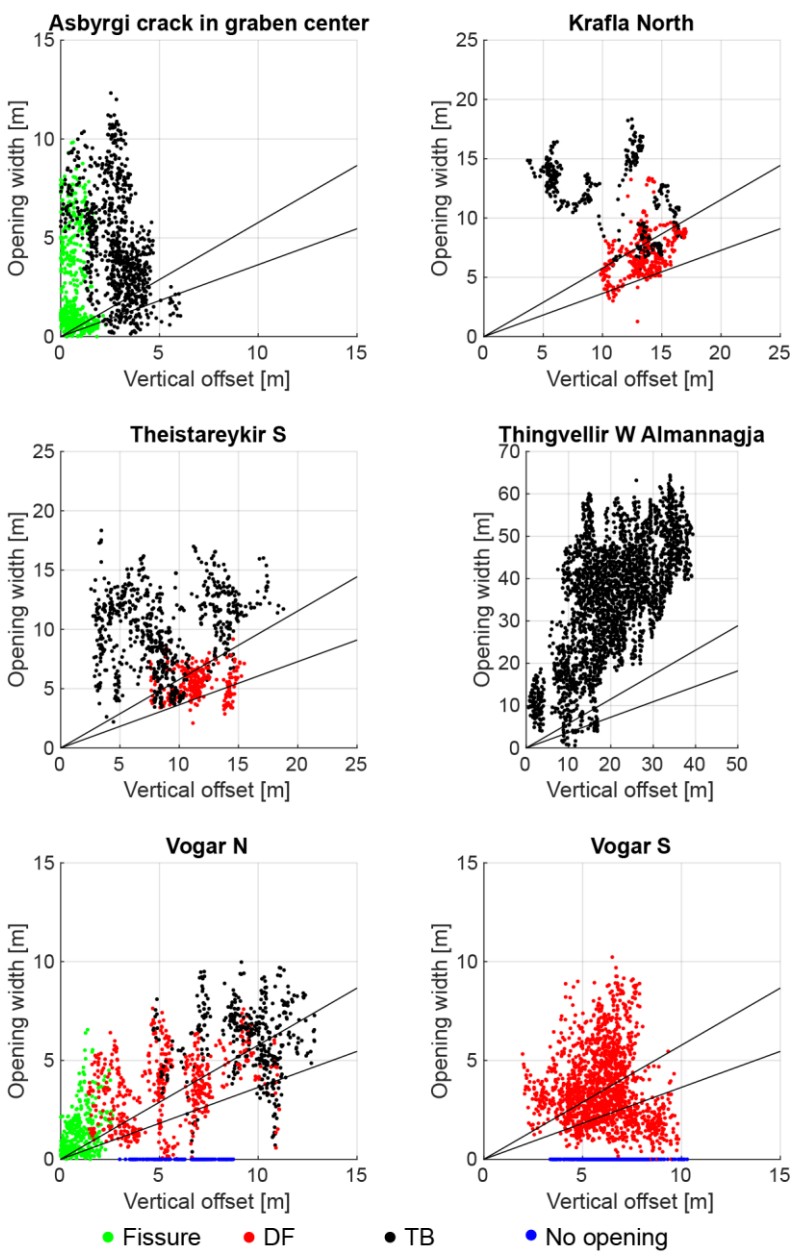

**Figure 20:** Plots of opening width against vertical offset. The black straight lines represent R for basement fault dips of 60° (upper one) and 70° (lower one).







**Figure 21:** Left: Cumulative plots of opening width plotted against the vertical offset, including all reviewed fractures. Right: Cumulative plots of opening width plotted against the vertical offset with data from Tentler and Mazzoli (2005) and Trippanera et al. (2015), excluding Thingvellir. The black straight lines represent R for basement fault dips of 60° and 70°.

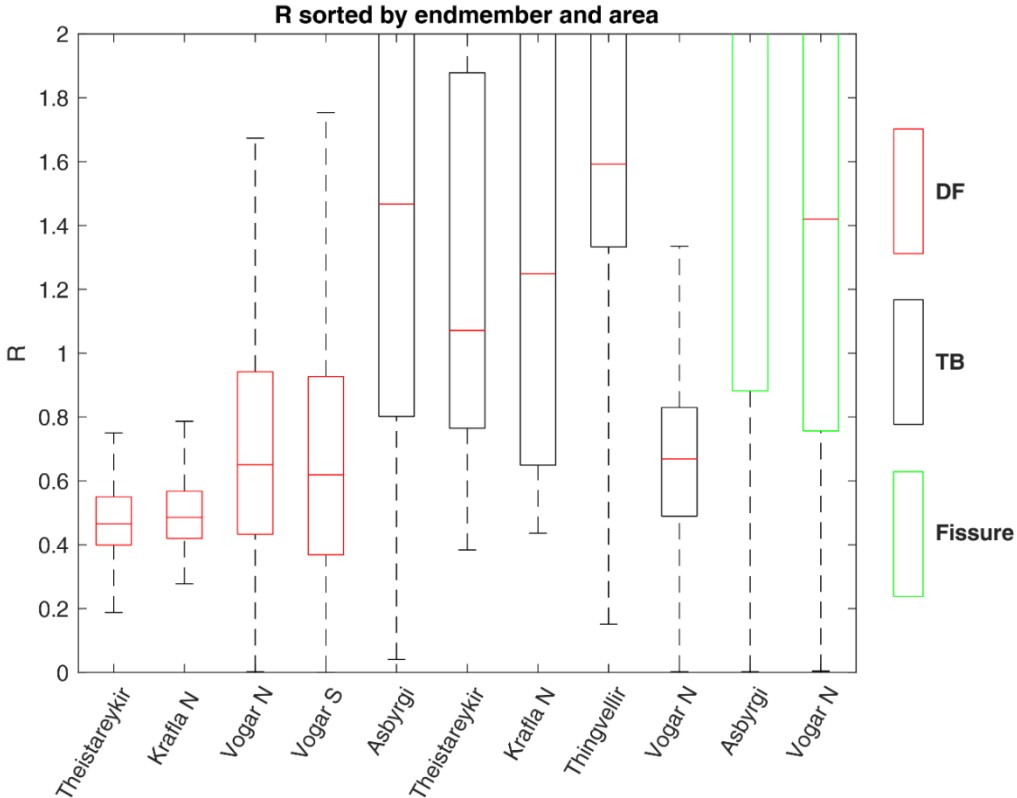

**Figure 22:** Boxplots of R sorted by endmember type and area. The y-axis of the graph has been cut at R = 2 to show the majority of values (O >> V will result in very large R). Outliers were removed. Maximum values of cut-off whiskers: Asbyrgi TB: 897.23, Theistareykir TB: 5.34, Krafla N: 4.02, Thingvellir N: 21.78, Asbyrgi fissures: 5499.11, Vogar N Fissures: 174.62. Dilatant faults with filled openings (O = 0) have been excluded, because R(O=0) = 0. Measurements with V = 0 have not been encountered, due to the surface roughness.

## 7. Data availability

A table containing the measurements of opening width and vertical offset and additional figures are provided in the supplement.

## 8. Author contributions

Christopher Weismüller acquired the data, did the processing and data extraction and prepared the manuscript with contribution from the co-authors. Janos L. Urai contributed to the conceptualization, discussion and interpretation of the data and acquired funding. Michael Kettermann acquired data and contributed to the processing, conceptualization and discussion. Christoph von Hagke and Klaus Reicherter contributed to the conceptualization and discussion of the manuscript and acquired funding.





## 9. Competing interests

The authors declare that they have no conflict of interest.

## 10. Acknowledgements

We would like to thank the park administration of the Thingvellir National Park for giving us the permission to use our
drone in the park outside of business hours and Daniele Trippanera for sharing his data with us. Further thanks go to the
Deutsche Forschungsgemeinschaft for funding this project. Project number 316167043. Project name: MDF: The structure
and evolution of near-surface massively dilatant faults. TanDEM-X WorldDEM™ data is provided by a DLR Science grant,
2017.

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
