# Peer review of "Structure of massively dilatant faults in Iceland: lessons learned from high resolution UAV data"

_Solid Earth, 2019_

## Referee Comment (RC1) · Anonymous Referee #1 · 30 Jul 2019

In the present paper, the authors studied the surficial expression of some normal faults in Iceland, mainly using UAV-derived digital surface models and orthomosaics. They classified them, based on the surficial expression, as well as they collected several quantitative measurements and provided dilation and vertical offset profiles; they also related vertical offset and horizontal dilation with the aid of field checks, to provide new findings on the above-cited topic. I generally appreciate this kind of study where plenty of data are provided and that present new approaches and technologies, and I recommend the paper for publication, but only after a major review with the aim of improving the structure of the manuscript, data presentation and to better highlight the results.

[Figure]

Introduction and discussion The introduction must be improved addressing in clear manner the methodology or the scientific problem presented in the paper. In the present form, there is a large list of cited literature but the subject of the paper is a bit vague. It is not clear if that is a test of a new methodology to study normal faults (e.g. UAV survey without GCPs) or if the aim is to present new findings on fault classification at the surface. Up to now, it looks somewhere in the middle. In addition, the discussion must be better addressed; the core of the paper is unclear in this chapter and it is difficult to appreciate the value of the new data and consequent results. I strongly recommend reorganizing this section after the introduction has been improved and thus the focus of the paper has been clarified. In the present form, it is hard to understand where the new findings are, regarding both the method and the scientific problem. Part of the scientific core of the paper seems to be presented in Section 1.2 that now belongs to the geological background. Geological background Sections 1.1 and 1.3 can be merged; they both describe the studied areas. Methods 2.2 Authors applied the areal Structure from Motion technique using a "border line" level of frontal and side overlap, without GCPs. This has surely affected the quality and the accuracy of the model, and must be discussed more in the paper, please do quantify the error. It also seems that the authors have not added any scale to the model/dense cloud. In addition, referencing them with lower resolution dataset could have also introduced errors. Results Sections 3 and 4 can be merged since in both of them results are presented. Conclusions This section must be better addressed in order to highlight new findings, after that the introduction and discussion sections have been both improved, as suggested above. Figures At a general level, the number of figures is too high. Some of them must be merged, especially when they are presenting the same type of data. 1. The caption can be shortened, eventually adding details in the figure. 2. North and scale are missing. Change the symbol for surveyed faults; star is often used to indicate earthquake epicenters.

---

## Referee Comment (RC2) · Anonymous Referee #2 · 11 Aug 2019

Dear Editor, The manuscript by WeismuÌĹller et al., is focused on studying the geometry of the Icelandic rift related faults by means of high-resolution images (mainly drone images). By using their very nice dataset, they aim at classifying the faults according to their geometry (e.g. presence or absence of a tilted block in the hanging wall) and the R value (ratio between the opening with and the vertical offset). The main outcome is that all different fault geometries are not due to a different origin (which is not clarified in the text) but represent structural endmembers of a continuum in the evolution of the faults or fissures. I suggest the publication of this manuscript after some minor revisions. Below are my suggestions.

[Figure]

Page 2, lines 8-9: I believe that in this case the "Campi Flegrei" example is not a very good one. Those you reported here- such as Icelandic and East African Rift - are the best examples of MDF that form on a rift, along the fissure swarm. As for the Hawaii example, I believe you are referring to the Koae Fault Zone on Kilauea that is also belonging to a huge rift zone affecting the whole volcano and related to the extension and dike intrusions related to the flank seaward motion. The fault observed on Campi Flegrei does not show such a clear relationship with a rift zone and most of deformation is mostly due to the caldera collapse and the interaction with the Appennines structural trend. Here I suggest to remove the Campi Flegrei citation. If you want to see a good example of MDF related to the caldera collapse (and not directly to the rifting activity) I suggest you to have a look at Askja caldera in Iceland (Trippanera et al., 2018, Bulletin of Volcanology).

Page 2, line 14: I think that here and elsewhere citations should be listed in temporal and not in alphabetical order.

Page 2, line 29: I believe that it would be probably better to invert the sequence of Fig. 1 and 2.

Page 2, line 30: "Maximum horizontal stresses..." Why you refer to the maximum horizontal stress instead of the minimum one as usual (that can be easily related to GPS vectors)?

Page 4, line 27: Please cite Fig. 2a, b at the end of the first sentence.

Page 5, line 2: I would remove the words "remote sensing" here. In this case "remote sensing" is clearly referred to satellite images but then later at lines 8 and 10 this is referred to aerial images, therefore it is a bit misleading.

Page 5, line 11: Gudmundsson et al., 1992 is about the 1991 Hekla eruption not the 2000 eruption. Please adjust the sentence accordingly.

Page 5, line 18: Did you use aerial images to make Sfm-DEMs? Please spell Sfm the

first time you use it in the manuscript.

Page 5, line 22: "...aid the mapping of faults and joints..." On this topic, I suggest to check also Trippanera et al., 2019 (Frontiers, Structural Geology and Tectonics).

Page 6, lines 3-4: "The photographs were sorted according to associated survey areas and reduced to only use sharp photographs with good image quality" This sentence could be deleted.

Page 6, line 16: "...significant error..." Could you indicate what is a significant error for you? (e.g. < 0.5 m)

Page 6, lines 25-26: "...With a mapping accuracy of a few mm in the DEM and ortho-mosaic at a 1:100 scale, the mapping error is in the same order of magnitude as our spatial resolution..." This is unclear. Please, revise it.

Page 6, line 27: "DF" You used MDF earlier. Please use MDF or spell it if you intend something different.

Page 7, sections 2.3.1 and 2.3.2 could be merged

Page 8, lines 17 to 19: from "The combination..." to "...during the processing." These lines should go in the "methods" section.

Page 8, lines 22 to 25: from "To simplify..." to "...above (Fig. 7-11)." These lines should go in the "captions" section.

Page 8, line 28: what do you mean for "maximum elevation difference"? Is it the difference between the highest and the lowest points? Please, make sure the reader does not understand that 44 m is the maximum fault throw.

Page 8, line 23 "scanline count" What do you mean with this?

Page 9, lines 4-5: how is the cross-section geometry in the segments i – ii – iii? Both footwall and hanging wall are flat (it seems like this from the DEM)?

Page 9, line 16: I believe you should describe the cross-section geometry of the fault before talking about TB vs opening. E.g. why not showing the variation of the tilt angle of the TB in the bottom diagram (Fig. 8) Do you have any idea about the relationship between the main fault and the fissures (in the DEM of Fig. 8) that seem to be oblique to the main fault?

Page 9, line 17: do you mean that the larger openings are associated with tilted hanging walls? If yes, I suggest to not use the word "slope" here and in line 18 but to directly refer to the tilt of the hanging wall (or TB).

Page 9, line 22-23: "Further trends of dipping surfaces are located on the hanging wall..." Do you mean the dip of the TB? It is not clear. It is better to describe the cross section geometry in one or more point, if different.

Page 9, line 26: "...bending ca. ...further north." How far are you from the intersection with the Husavik – Flatey fault?

Page 10, lines 2-3: "since a horizontal hanging wall is not covered in the north and the south at the TBs" Unclear

Page 11, lines 8-9: Probably between Vogar and the other rift systems, there should be a difference in the opening direction. Do you have any info about this?

Page 11, line 12: You should recall figure 12a before 12b in the text.

Page 15, line 26: "The difference is a result of our definition for the cutoff of vertical offset (2 m)" This is a bit unclear.

Page 16, lines 3 ("...mapping error...") and 10 ("...vertical offset are underestimated..."): Can you quantify the range of the errors for the vertical offset?

Page 16, lines 11 to 13: "Overlaps of DF and TB....no clear boundary is visible." I believe that a block with a dip value «5° cannot be classified as a TB in any case.

Perhaps, there is an opening (or R value) threshold after which only TB develops and below this threshold one can have both DF and TB.

Page 16, lines 29-30: "This is because vertical offset is less influenced by surface 30 structures as the opening width and measured outside the influence area of TB." Not clear

Page 17, line 23: What is the meaning of "al.st."?

Page 17, line 32: Do you mean the location of drone surveys along the fault are spotted?

Figure 1: It could be better if you indicate the footwall and the hanging wall in each figure.

Figure 8 and elsewhere: I suggest to indicate the foot wall and the hanging wall in the DEM, for a more immediate understanding of the figure.

Figures 13 to 16: it could be nice to have also a topographic profile across the main features.

Figure 22: It would be useful to add a background reference (e.g. a light gray box) for the expected R values

―――――――――――――――――――――

---

## Author Comment (AC1) · 11 Sep 2019

Reply to 'Comments' by Anonymous Referee #1:

Anonymous Referee #1

In the present paper, the authors studied the surficial expression of some normal faults in Iceland, mainly using UAV-derived digital surface models and orthomosaics. They classified them, based on the surficial expression, as well as they collected several quantitative measurements and provided dilation and vertical offset profiles; they also related vertical offset and horizontal dilation with the aid of field checks, to provide

new findings on the above-cited topic. I generally appreciate this kind of study where plenty of data are provided and that present new approaches and technologies, and I recommend the paper for publication, but only after a major review with the aim of improving the structure of the manuscript, data presentation and to better highlight the results.

Reply: We thank Reviewer #1 for the comments on the manuscript. A main criticism was that the objective was not clearly stated, and some restructuring of the manuscript was in order. In the new version of the manuscript we have taken special care to clarify the scope of the study, and restructured the manuscript particularly in section 1, as requested. Suggestions to merge parts of the results and interpretation were not accommodated, but we consider this mostly a matter of taste how to lay out the manuscript. Detailed comments on the individual points raised by the reviewer are provided below.

Introduction and discussion The introduction must be improved addressing in clear manner the methodology or the scientific problem presented in the paper. In the present form, there is a large list of cited literature but the subject of the paper is a bit vague. It is not clear if that is a test of a new methodology to study normal faults (e.g. UAV survey without GCPs) or if the aim is to present new findings on fault classification at the surface. Up to now, it looks somewhere in the middle.

Reply: We have used a, to our requirements modified, version of a recent, but well established technique to create digital elevation models (DEM) from unmanned aerial vehicle (UAV) photographs. Based on these data, we developed a workflow to extract large amounts of measurements in high resolution, which enabled us to introduce a new classification scheme for the faults at the surface in combination with our field observations. Therefore, the manuscript indeed deals with both aspects: The UAV-DEM are the methodology we used to acquire the base for our geometrical analysis of the faults and fractures, which is the main aspect of the manuscript. To clarify this, we modified the final paragraph of the introduction to make the goal of the paper, using high-res

UAV data over kilometer scale faults to improve the surface fault classification beyond simple geometric observations. We agree with the reviewer that this modification will make the scope of the study clearer to the reader.

In addition, the discussion must be better addressed; the core of the paper is unclear in this chapter and it is difficult to appreciate the value of the new data and consequent results. I strongly recommend reorganizing this section after the introduction has been improved and thus the focus of the paper has been clarified. In the present form, it is hard to understand where the new findings are, regarding both the method and the scientific problem.

Reply: Our discussing starts with a short recap of our methods and the therewith achieved resolutions, before we put our methods in respect to the literature. We subsequently introduce the parameter R, which will be used to further discuss our measurements and findings in the rest of the discussion. In the following section, we analyze distinctive features in our models and measurements to verify the applicability and special cases of our classification. This is done by first analyzing these features in the single models that have been introduced, before we include further data from ourselves and literature for a more general discussion. In that part, we test and describe our classification scheme more universally, leading to the conclusion that the different endmembers are part of a larger continuum. We are convinced, that this organization is a proper way to discuss our large amount of data in respect to the findings from our measurements, as the sections are not interchangeable but build up on the discussions in the preceding section. Therefore, we decided not to restructure the discussion. However, we hope that with the changes in the introduction and restructuring in the data presentation the scope of the paper as well as what are new findings is now clear.

Part of the scientific core of the paper seems to be presented in Section 1.2 that now belongs to the geological background. Geological background Sections 1.1 and 1.3 can be merged; they both describe the studied areas.

[Figure]

Reply: Section 1.2 introduces the reader in the topic of massively dilatant faults (MDF) and provides a summary of existing literature dealing with the formation and geometry of MDF, before the last lines put this manuscript in perspective to existing literature. This belongs rather in the introduction part and is not supposed to be the scientific core of this paper, as we are only dealing with one aspect out of the large variety of aspects of MDF described in this section. Section 1.1 is a broad introduction to the regional setting, while section 1.3 deals with the setting of our study areas in detail. We agree that these section can be merged. To highlight and underline the introductory character of the MDF section (1.2) we decided to move it up as new section 1.1, followed by a new section 1.2 which consists of the merged sections dealing with the regional and detailed settings of our study areas.

Methods 2.2 Authors applied the areal Structure from Motion technique using a "border line" level of frontal and side overlap, without GCPs. This has surely affected the quality and the accuracy of the model, and must be discussed more in the paper, please do quantify the error.

Reply: The general assumption that we used a "border line" overlap for our model is not correct. The values referred to represent the minimum values we worked with, mostly in the distant regions of our models. The important areas including the mapped fractures have been covered with higher overlaps. We have made this clearer now in the text. The impact of not using GCP in our data and how we dealt with this circumstance, also in terms of quality check, is already explained later in the methods (e.g, p.6 line 8 ff.), also with an error estimation derived from the onboard GPS accuracy, and additionally in the "ground truthing and field observations" section. We would like to point out, that this manuscript does not aim to be a methodological paper about UAV-Sfm and DEM generation, as we have just slightly adjusted an established methodology.

It also seems that the authors have not added any scale to the model/dense cloud.

Reply: We did use the UAV onboard GPS, which serves as scale during the processing.

[Figure]

Using the software (Agisoft Photoscan), it is not possible to create and export a DEM without scale or reference of the image-chunk. Quality checking of the model scale has been briefly explained in the ground truthing section. We have moved the paragraph to section 2.2.

In addition, referencing them with lower resolution dataset could have also introduced errors.

Reply: We did not reference our models with a lower resolution dataset, nor do we state doing so. We used the lower resolution TanDEM-X data to compare our models to in order to quality check with regional morphology, e.g. to identify local slopes of the surface. By doing so, we were able to rule out an artificial tilt of our models, which is a possible artifact when no GCP's are used. This is described in e.g. section 2.1.

Results Sections 3 and 4 can be merged since in both of them results are presented.

Reply: Section 3 shows pure results which are addressed as objectively as possible. Section 4 is the interpretation of the results shown in 3 and the structures are already classified as different endmembers/fracture types, according to our interpretation of the situation. Thus, while both being similar, section 3 aims to objectively represent the data, while section 4 is our subjective interpretation. Therefore, we would like to keep these sections separate to draw a clear line for the reader between results and interpretation.

Conclusions This section must be better addressed in order to highlight new findings, after that the introduction and discussion sections have been both improved, as suggested above.

Reply: We restructured the conclusions to clearer separate new observations and the most important conclusions.

Figures At a general level, the number of figures is too high. Some of them must be merged, especially when they are presenting the same type of data.

Reply: One of the fundamentals of this manuscript is to present and introduce the reader to our high-resolution dataset from a comprehensive study. Therefore, we deem the number of figures as necessary to provide the reader with a sufficient amount of information to make our interpretation and argumentations as transparent and reproducible as possible for the audience. To avoid repetitions and things of minor importance in the manuscript itself, an addition supplement with further data is already provided. We decline the suggestion to merge further figures, as we deem the perceptibility of details in our figures, when dealing with high resolution data, as one very important aspect. Merging the figures would decrease the resolution and therefore undermine this concept, especially when the figures are viewed in paper form and not on a screen that allows zooming in and out. We thereby merged already as many figures as possible and kept the figure count as low as we could, without affecting the quality of our data presentation.

1. The caption can be shortened, eventually adding details in the figure.

Reply: We agree. The caption of Fig. 1 has been shortened in favor of more details added in the figure itself, namely the hummocks, hanging wall and footwall, lava flows and possible fillings of the faults.

2. North and scale are missing. Change the symbol for surveyed faults; star is often used to indicate earthquake epicenters.

Reply: While we initially determined the existing coordinate system as sufficient, we agree that North and a scale make the figure easier and faster to understand. North and a scale have been added in each subfigure. To avoid confusion with epicenters, we have replaced the stars with rhombs.

---

## Author Comment (AC2) · 11 Sep 2019

Dear Editor, the manuscript by Weismüller et al., is focused on studying the geometry of the Icelandic rift related faults by means of high-resolution images (mainly drone images). By using their very nice dataset, they aim at classifying the faults according to their geometry (e.g. presence or absence of a tilted block in the hanging wall) and the R value (ratio between the opening with and the vertical offset). The main outcome is that all different fault geometries are not due to a different origin (which is not clarified in the text) but represent structural endmembers of a continuum in the evolution of

the faults or fissures. I suggest the publication of this manuscript after some minor revisions. Below are my suggestions.

Reply: We thank Reviewer #2 for his careful and detailed comments, almost all of which have been accommodated in the new version of the manuscript.

Page 2, lines 8-9: I believe that in this case the "Campi Flegrei" example is not a very good one. Those you reported here- such as Icelandic and East African Rift - are the best examples of MDF that form on a rift, along the fissure swarm. As for the Hawaii example, I believe you are referring to the Koae Fault Zone on Kilauea that is also belonging to a huge rift zone affecting the whole volcano and related to the extension and dike intrusions related to the flank seaward motion. The fault observed on Campi Flegrei does not show such a clear relationship with a rift zone and most of deformation is mostly due to the caldera collapse and the interaction with the Appennines structural trend. Here I suggest to remove the Campi Flegrei citation. If you want to see a good example of MDF related to the caldera collapse (and not directly to the rifting activity) I suggest you to have a look at Askja caldera in Iceland (Trippanera et al., 2018, Bulletin of Volcanology).

Reply: This is a really good point we agree with. We have removed the Campi Flegrei citation accordingly. Also, thanks for pointing us to Trippanera et al., 2018, this is an interesting publication and might be useful for us in the future.

Page 2, line 14: I think that here and elsewhere citations should be listed in temporal and not in alphabetical order.

Reply: We have used the Zotero plugin with the CSL Style provided by Copernicus on their website for the manuscript preparation. Our reference style should be in line with the Copernicus publication guidelines for authors: https://www.solid-earth.net/for_authors/manuscript_preparation.html. Therefore, we did not change the order of the authors.

Page 2, line 29: I believe that it would be probably better to invert the sequence of Fig. 1 and 2.

Reply: We agree with this point. However, we have merged sections 1.1 and 1.3 according to the other review. Fig. 1 showing the MDF remains on position 1, as we are now explaining MDF in general first, followed by the geological background section with the map in Fig. 2.

Page 2, line 30: "Maximum horizontal stresses..." Why you refer to the maximum horizontal stress instead of the minimum one as usual (that can be easily related to GPS vectors)?

Reply: We used the max. horizontal stresses as presented in Ziegler et al. 2016, because the orientation matches the faults strike and we did not deal with GPS vectors. We now adjusted it to Minimum horizontal stresses, as suggested, because we believe this makes it easier for the reader to understand the setting. The orientations in the following sentences have been adjusting accordingly.

Page 4, line 27: Please cite Fig. 2a, b at the end of the first sentence.

Reply: We have cited the figure at the end of the sentence.

Page 5, line 2: I would remove the words "remote sensing" here. In this case "remote sensing" is clearly referred to satellite images but then later at lines 8 and 10 this is referred to aerial images, therefore it is a bit misleading.

Reply: We agree and removed "remote sensing" from the sentence.

Page 5, line 11: Gudmundsson et al., 1992 is about the 1991 Hekla eruption not the 2000 eruption. Please adjust the sentence accordingly.

Reply: Absolutely, we have adjusted the sentence by also referring to the 1991 eruption.

Page 5, line 18: Did you use aerial images to make Sfm-DEMs? Please spell Sfm the

first time you use it in the manuscript.

Reply: Yes, we used aerial images we have captured with our unmanned aerial vehicles. Sfm is now written out.

Page 5, line 22: "...aid the mapping of faults and joints..." On this topic, I suggest to check also Trippanera et al., 2019 (Frontiers, Structural Geology and Tectonics).

Reply: Thanks for pointing us towards this publication. It was just published after we submitted the initial manuscript. We have added Trippanera et al. 2019 to our references.

Page 6, lines 3-4: "The photographs were sorted according to associated survey areas and reduced to only use sharp photographs with good image quality" This sentence could be deleted.

Reply: As the sentence explains a self-evident procedure, we have removed it.

Page 6, line 16: "...significant error..." Could you indicate what is a significant error for you? (e.g. < 0.5 m)

Reply: We were referring to a significant error in the sense of an error we can identify with the given datasets with different resolutions (Our UAV-DEMS and TanDEM-X). As our resolution is much finer, we were not able to identify a horizontal mismatch with TanDEM-X or other satellite data, therefore we assume the horizontal error is in the same order of magnitude as our onboard GPS receiver. This is addressed in the preceding sentences. We have adjusted the sentence and added a following one to make this clearer in the manuscript.

Page 6, lines 25-26: "...With a mapping accuracy of a few mm in the DEM and orthomosaic at a 1:100 scale, the mapping error is in the same order of magnitude as our spatial resolution..." This is unclear. Please, revise it.

Reply: We have rewritten the sentence to make clear that we are referring to the man-

ual mapping accuracy in our UAV-data in GIS, where we are interpreting the fractures at a mm scale on the screen which represents a true scale of a few centimeters in the outcrop.

Page 6, line 27: "DF" You used MDF earlier. Please use MDF or spell it if you intend something different.

Reply: Dilatant Faults (DF) is meant to be a general term of faults with a (also small) dilatancy, while MDF is referring to DF that have a large dilatancy in the range of several tens of meters. DF is now written out in the sentence.

Page 7, sections 2.3.1 and 2.3.2 could be merged

Reply: We agree and have merged the sections into one, as they are dealing with similar methods.

Page 8, lines 17 to 19: from "The combination..." to "...during the processing." These lines should go in the "methods" section.

Reply: We agree, the lines have been moved to the methods.

Page 8, lines 22 to 25: from "To simplify..." to "...above (Fig. 7-11)." These lines should go in the "captions" section.

Reply: We agree, the lines have been moved to the caption of Fig. 7.

Page 8, line 28: what do you mean for "maximum elevation difference"? Is it the difference between the highest and the lowest points? Please, make sure the reader does not understand that 44 m is the maximum fault throw.

Reply: We are indeed referring to the highest and the lowest points in the model. This is also addressed in section 2.2, we have added a sentence to make it clear as well in this line.

Page 8, line 23 "scanline count" What do you mean with this?

Reply: The scanlines to measure the opening width have been created in 1 m intervals. Therefore, the counter of each scanline also represents its position along strike in m. We have removed "scanline count" in this line, as it is rather confusing instead of adding valuable information here.

Page 9, lines 4-5: how is the cross-section geometry in the segments i – ii – iii? Both footwall and hanging wall are flat (it seems like this from the DEM)?

Reply: The surface is rather smooth and the DEM not very broad in this direction. Due to the larger N-S elevation difference, the E-W slope is not well visible. This issue is addressed later in Section 4 Asbyrgi, where we used the TanDEM-X tiles to identify this tilt and aid our interpretation.

Page 9, line 16: I believe you should describe the cross-section geometry of the fault before talking about TB vs opening. E.g. why not showing the variation of the tilt angle of the TB in the bottom diagram (Fig. 8) Do you have any idea about the relationship between the main fault and the fissures (in the DEM of Fig. 8) that seem to be oblique to the main fault?

Reply: We agree, that it is better to describe the cross section geometry first. Tilt angles of the TB are not reviewed in detail, as they are part of Kettermann et al. 2019. The fissures on the hanging wall are most likely related to diking, and might be eruptive fissures linked to the young lava flow in the south. But we cannot verify this at the current state, as our focus in field was the main fault. We have added a line about the hanging wall geometry first.

Page 9, line 17: do you mean that the larger openings are associated with tilted hanging walls? If yes, I suggest to not use the word "slope" here and in line 18 but to directly refer to the tilt of the hanging wall (or TB).

Reply: Yes, this is what we meant. We did not use the terminology TB here, to not interpret the structures in the result section, but rather give an objective description

of the DEM. We have replaced the "slopes" with "tilted hanging walls", and added the assumption, that these are most likely TB.

Page 9, line 22-23: "Further trends of dipping surfaces are located on the hanging wall..." Do you mean the dip of the TB? It is not clear. It is better to describe the cross section geometry in one or more point, if different.

Reply: We are referring to the TB. The sentenced has been rephrased to make this clearer.

Page 9, line 26: "...bending ca. ...further north." How far are you from the intersection with the Husavik – Flatey fault?

Reply: Ca. 10 km south of the southern part of the Husavik-Flatey transform zone as mapped in Bonali et al. 2019. Later in section 4 we interpret the decreasing opening-width observed at the fault segment as result from obliquity, possibly due to the proximity to the Husavik-Flatey transform zone.

Page 10, lines 2-3: "since a horizontal hanging wall is not covered in the north and the south at the TBs" Unclear

Reply: We intended to describe that measurements of vertical offset can only be performed on top of the tilted hanging walls in those regions, and not the flat hanging wall surface which is without the influence of TB. Therefore, measurements of vertical offset will be underestimated. We changed the sentence to make this clearer.

Page 11, lines 8-9: Probably between Vogar and the other rift systems, there should be a difference in the opening direction. Do you have any info about this?

Reply: This is correct, as Vogar is located on the Reykjanes Peninsula where the stress field has a different orientation than e.g. the North Volcanic Zone. However, for our classification the opening direction is less important, as we are focusing on the general fracture geometries and not regional tectonics. The influence of stress field orientation and obliquity has previously been addressed in von Hagke et al., 2019.

Page 11, line 12: You should recall figure 12a before 12b in the text.

Reply: Of course. We have changed the fissures section with the dilatant faults section, now (a) is mentioned before (b). However, (c) is now mentioned before (b) in the text, but we would like to keep this order as the two special types of Dilitant Faults are easier to compare when on top of each other in the figure due to their similarities.

Page 15, line 26: "The difference is a result of our definition for the cutoff of vertical offset (2 m)" This is a bit unclear.

Reply: We are referring to the different endmember-classification of the measurements, which is caused by our definition of fissures. We have added a few lines to explain this with more in detail.

Page 16, lines 3 ("...mapping error...") and 10 ("...vertical offset are underestimated..."): Can you quantify the range of the errors for the vertical offset?

Reply: The first part (mapping error) is dealing with the influence of the vegetation and soil cover in the measurements in the DEM. These circumstances are addressed previously and more detailed in section 2.3.2 and Fig. 5. We are now referring the reader towards these sections, to make it more clear. The error in vertical offset in the Theistareykir S DEM is now estimated at 5 m and further explained. The estimation is derived from the measurements taken on the central part of the hanging wall without the influence of the surrounding TBs.

Page 16, lines 11 to 13: "Overlaps of DF and TB....no clear boundary is visible." I believe that a block with a dip value Âń5_ cannot be classified as a TB in any case. Perhaps, there is an opening (or R value) threshold after which only TB develops and below this threshold one can have both DF and TB.

Reply: Indeed, a tilt less than ca. $5°$ is too subtle to be reliably classified as a TB. For young faults with small vertical offsets, the TB might be immature and therefore only slightly tilted, until the rotation becomes more apparent with accumulating vertical

offset. The transition from tilted hanging wall to the horizontal can be very smooth, resulting in the overlapping zone of the two endmembers. We have added a short example at the end of the sentence to make this more comprehensible for the reader.

Page 16, lines 29-30: "This is because vertical offset is less influenced by surface structures as the opening width and measured outside the influence area of TB." Not clear

Reply: The vertical offset is mainly affected by the geometry of the basement fault, while the opening width is heavily altered by e.g. erosion. We have added a more detailed explanation in the following lines.

Page 17, line 23: What is the meaning of "al.st."?

Reply: "Along strike", initially introduced in the results sections. We have now replaced the abbreviation with the written out words in this line, as the abbreviation was previously defined for use in the results section, where it had to be used more frequently.

Page 17, line 32: Do you mean the location of drone surveys along the fault are spotted?

Reply: No, this line is about the measurements in the DEM. They are only taken at single points within a defined interval of 1 m, therefore, transitions smaller than this interval appear as jumps in our plot instead of continuous transitions. This is further explained in Supplement 6 as referred to in the prior lines. We have added a sentence to shorty explain it in this section as well.

Figure 1: It could be better if you indicate the footwall and the hanging wall in each figure.

Reply: This is a good suggestion, we have added indicators for the hanging wall and footwall in each image.

Figure 8 and elsewhere: I suggest to indicate the foot wall and the hanging wall in the

[Figure]

DEM, for a more immediate understanding of the figure.

Reply: Hanging wall and footwall are already indicated by the color code of the elevation measurements, as explained in the respective legends of the figures.

Figures 13 to 16: it could be nice to have also a topographic profile across the main features.

Reply: We agree, that this would be a nice addition. However, this would further contribute to our already high number of figures, which has been criticized by the other reviewer. Therefore, we abstain from adding the profiles, as they can also be inferred by the reader from the color coded elevations in the figures.

Figure 22: It would be useful to add a background reference (e.g. a light gray box) for the expected R values.

Reply: This is a good suggestion how to increase the information content of the figure. We have added a grey box indicating the R(70)-R(60) range.